

# A global dataset of spatiotemporally seamless daily mean land surface temperatures: generation, validation, and analysis

Falu Hong[1], Wenfeng Zhan[1, 2*], Frank-M. Göttsche[3], Zihan Liu[1], Pan Dong[1], Huyan Fu[1], Fan Huang[1], and Xiaodong Zhang[4]

[1]Jiangsu Provincial Key Laboratory of Geographic Information Science and Technology, International Institute for Earth System Science, Nanjing University, Nanjing, Jiangsu 210023, China
[2]Jiangsu Center for Collaborative Innovation in Geographical Information Resource Development and Application, Nanjing 210023, China
[3]Karlsruhe Institute of Technology (KIT), Hermann-von-Helmholtz-Platz 1, 76344 Eggenstein-Leopoldshafen, Germany
[4]Shanghai Spaceflight Institute of TT&C and Telecommunication, Shanghai, 201109, China

*Correspondence to*: Wenfeng Zhan (zhanwenfeng@nju.edu.cn)

**Abstract.** Daily mean land surface temperatures (LSTs) acquired from polar-orbiters are crucial for various applications such as global and regional climate change analysis. However, thermal sensors from polar-orbiters can only sample the surface effectively with very limited times per day under cloud-free conditions. These limitations have produced a systematic sampling bias ($\Delta T_{sb}$) on the daily mean LST ($T_{dm}$) estimated with the traditional method, which uses the averages of clear-sky LST observations directly as the $T_{dm}$. Several methods have been proposed for the estimation of the $T_{dm}$, yet they become less capable of generating spatiotemporally seamless $T_{dm}$ across the globe. Based on MODIS and reanalysis data, here we proposed an improved annual and diurnal temperature cycle-based framework (termed the IADTC framework) to generate global spatiotemporally seamless $T_{dm}$ products ranging from 2003 to 2019 (named as the GADTC products). The validations show that the IADTC framework reduces the systematic $\Delta T_{sb}$ significantly. When validated only with *in situ* data, the assessments show that the mean absolute errors (MAEs) of the IADTC framework are 1.4 K and 1.1 K for SURFRAD and FLUXNET data, respectively; and the mean biases are both close to zero. Direct comparisons between the GADTC products and *in situ* measurements indicate that the MAEs are 2.2 K and 3.1 K for the SURFRAD and FLUXNET datasets, respectively; and the mean biases are −1.6 K and −1.5 K for these two datasets, respectively. By taking the GADTC products as references, further analysis reveals that the $T_{dm}$ estimated with the traditional averaging method yields a positive systematic $\Delta T_{sb}$ of greater than 2.0 K in low- and mid-latitude regions while of a relatively small value in high-latitude regions. Although the global mean LST trend (2003 to 2019) calculated with the traditional method and the IADTC framework is relatively close (both between 0.025 to 0.029 K/year), regional discrepancies in LST trend does occur – the pixel-based MAE in LST trend between these two methods reaches 0.012 K/year. We consider the IADTC framework can guide the further optimization of $T_{dm}$ estimation across the globe; and the generated GADTC products should be valuable in various applications such as global and regional warming analysis. The GADTC products are freely available at https://doi.org/10.5281/zenodo.6287052 (Hong et al., 2022).



## 1 Introduction

Land surface temperature (LST) is one of the most important variables of land-atmosphere interaction (Jin and Dickinson,
2010). Currently, satellite thermal remote sensing provides the only way to obtain long-term and regular LST over extensive
areas. The archived long-term satellite-derived LST datasets have been widely used in various fields such as land cover
change detection (Lambin and Ehrlich, 1997; Muro et al., 2018), radiation flux simulation (Alcântara et al., 2010; Anderson
et al., 2007), drought monitoring (Karnieli et al., 2010; Mildrexler et al., 2017), vegetation change analysis (Julien and
Sobrino, 2009; Julien et al., 2006; Still et al., 2019), permafrost thawing monitoring (Westermann et al., 2011), and global
LST trend investigation (Jin, 2004; Jin and Dickinson, 2002; Yan et al., 2020).

According to the satellite on-board duration and spatiotemporal resolution (Tomlinson et al., 2011), satellite-derived
LST products used for long-term time-series analysis can be divided into two categories: (1) the LSTs obtained from high-
orbit geostationary satellite sensors with a coarse spatial resolution (3 – 5 km), e.g., the MSG-SEVIRI (the Spinning
Enhanced Visible and Infrared Imager onboard Meteosat Second Generation) and GOES (Geostationary Operational
Environmental Satellite), and (2) the LSTs from low-orbit polar-orbiting satellite sensors. The second category of satellite
sensors can be further divided into (1) the narrow-swath polar-orbiting satellite sensors with a relatively high spatial
resolution (around 100 m), e.g., Landsat and ASTER (Advanced Spaceborne Thermal Emission and Reflection Radiometer)
and (2) the polar-orbiting satellite sensors with a moderate spatial resolution (around 1 km), e.g., AVHRR (Advanced Very
High-Resolution Radiometer), SLSTR (Sea and Land Surface Temperature Radiometer), and MODIS (Moderate Resolution
Imaging Spectroradiometer).

The geostationary satellite thermal sensors are characterized by a very high temporal resolution (1 hour or finer).
However, they are relatively difficult to provide global consistent LST products due to the limited coverage of a single
geostationary satellite and the systematic errors among different satellites (Freitas et al., 2013). The Landsat (or similar
polar-orbiters) has been providing thermal observations since the 1980s, but the relatively long revisiting period (e.g., 16-day
for Landsat) makes it challenging to capture the daily and hourly continuous LST dynamics (Fu and Weng, 2016). By
contrast, wide-swath polar-orbiting sensors (e.g., MODIS) can sample the earth surface at least twice a day with a relatively
high spatial resolution (around 1.0 km). The feature makes the MODIS-like sensors overcome the limitations of the Landsat-
like satellites (with a long revisiting period) and geostationary satellite sensors (with a restricted global coverage). Therefore,
the LSTs obtained from wide-swath polar-orbiting sensors (e.g., MODIS and AVHRR) have been widely used in capturing
the long-term global LST dynamics (Sobrino et al., 2020a; Mildrexler et al., 2011). Among these, the MODIS LST data have
been used the most (Eleftheriou et al., 2018; Fu, 2019; Heck et al., 2019; Potter and Coppernoll-Houston, 2019; Quan et al.,
2016; Sobrino et al., 2020a; Yan et al., 2020; Zhao et al., 2019; Zhao et al., 2021). This is mainly because, especially when
compared with the AVHRR data, (1) MODIS LST observations are less affected by the orbit drift effect (Julien and Sobrino,
2012; Latifovic et al., 2012; Ma et al., 2020; Gutman, 1999); (2) the MODIS LST products can offer more details about the
diurnal LST dynamics with four observations per day (Crosson et al., 2012; Hong et al., 2018); and (3) the MODIS LST



retrieval algorithm has been with continuous improvement and the associated LSTs products are comparably more mature and have been extensively validated (Duan et al., 2019; Wan, 2014).

However, most previous studies employed temporally aggregated results (8-day or monthly mean) of instantaneous cloud-free LSTs for long-term LST time series analysis (Mao et al., 2017; Sobrino et al., 2020a; Sobrino et al., 2020b; Xing
et al., 2021), instead of continuous daily mean LST (termed as $T_{dm}$) on a day-to-day basis. Compared with the continuous daily $T_{dm}$, temporally aggregated results of instantaneous cloud-free LSTs lack the information of under-cloud thermal observations and insufficiently sample the LST diurnal dynamics (Ermida et al., 2019; Hu et al., 2020; Westermann et al., 2012). Such a direct temporal aggregation approach can produce a systematic sampling bias (termed as $\Delta T_{sb}$) (Hong et al., 2021), which affects the accuracy of $T_{dm}$ directly and the associated trend analysis indirectly (Zhou and Wang, 2016). To
estimate accurate $T_{dm}$, Hong et al. (2021) designed the ADTC-based framework that combines an annual temperature cycle (ATC) model and a diurnal temperature cycle (DTC) model. Based on the MODIS LST product and some auxiliary data such as the reanalysis data, the ADTC-based framework first uses an ATC model to reconstruct the instantaneous under-cloud LSTs and then simulates the diurnal LST dynamics with a four-parameter DTC model to solve the issue of under-sampling with only four observations per day. Validations showed that the ADTC-based framework can reduce the $\Delta T_{sb}$
significantly and produce the spatiotemporally seamless $T_{dm}$ (Hong et al., 2021).

However, the original ADTC-based framework (termed the OADTC framework) has been tested only over a relatively small region. In other words, the performance of the OADTC framework over complicated situations across global land surfaces has not been studied. Currently a global spatiotemporally seamless daily mean LST product is still unavailable to the satellite thermal remote sensing community; furthermore, the spatial distribution of $\Delta T_{sb}$ and its impact on the LST trend
over global land surfaces also remains unclear. There are two further limitations when applying the OADTC framework to the actual generation of global seamless $T_{dm}$: (1) the selected ATC model in the OADTC framework uses a single sinusoidal function to describe the intra-annual variation of solar radiation, which becomes less suitable for equatorial and polar regions (Liu et al., 2019b); (2) the used DTC model may fail around sunrise with no-solution or extreme solution, and cause an underestimation and even outliers of the daily mean LST (Hong et al., 2021; Hu et al., 2020).

Facing these issues, this study intends to formulate an improved version of the original ADTC-based framework (hereafter termed the IADTC framework) by using an advanced multi-type ATC model as well as a DTC model optimized for estimating $T_{dm}$. With the IADTC framework, we then generate a global spatiotemporally seamless 0.5-degree $T_{dm}$ product (termed the GADTC product, refer to Section 3.1 for the detailed description) for the period from 2003 to 2019. Based on the GADTC product, we then analyze the global spatial distribution of $\Delta T_{sb}$ as well as LST trends, which are compared with
those obtained with the traditional method. We consider the IADTC framework and the associated GADTC product should be useful for various applications such as analysis of global climate change and assessment of reanalysis data.

## 2 Datasets

The MODIS LST products and MERRA2 (the Modern-Era Retrospective analysis for Research and Applications version 2) reanalysis dataset were required as input data for the IADTC framework. We also employed *in situ* LST measurements from
the SURFRAD and FLUXNET to validate the IADTC framework and the GADTC product.

### 2.1 MODIS LST products

The MODIS LST products, including both the MOD11C1 and MYD11C1 LST products in Collection 6 from 2003 to 2019 (available at https://ladsweb.nascom.nasa.gov/), were used to help the generation of $T_{dm}$. The MODIS LSTs were retrieved with a refined generalized split-window algorithm, and their accuracies are mostly within 1.0 K over homogeneous surfaces
(Duan et al., 2019; Wan, 2014). The MOD11C1 and MYD11C1 LST products cover the global land surfaces four times per day with a spatial resolution of 0.05 degree. At low- and middle-latitude regions, MOD11C1 LSTs are obtained around 10:30 and 22:30 (local solar time), and MYD11C1 LSTs are around 01:30 and 13:30 (local solar time) with a time interval of around 1.5 hours. At high-latitude regions, due to the convergence of satellite orbit (Fig. A1), the overpassing times possess a significant shift from those at low- and middle-latitude regions (Østby et al., 2014). More details on the time shift and its
impact on the estimation of $T_{dm}$ with the IADTC framework are provided in Sections 3.1.3 and 5.2.

### 2.2 Reanalysis data

Surface air temperatures (SATs) are used to drive the ATC model for the reconstruction of under-cloud LSTs (see Section 3.1). We employed the SATs from MERRA2 reanalysis dataset (the specific collection name is inst1_2d_lfo_Nx, obtained from https://disc.gsfc.nasa.gov/datasets/M2I1NXLFO_V5.12.4/summary) from 2003 to 2019 (Gelaro et al., 2017; GMAO
2015). The spatial and temporal resolutions of these reanalysis SAT data are 0.5×0.625 degrees and 1 hour, respectively.

### 2.3 In situ data

The *in situ* LST measurements from 133 globally distributed stations (Fig. 1) were used to validate the IADTC framework at site level (see Section 3.2.1) as well as to evaluate the GADTC product (see Section 3.2.2). They include seven SURFRAD (Surface Radiation Budget Network) sites (Augustine et al., 2000) and 126 FLUXNET sites from FLUXNET2015 datasets
(Pastorello et al., 2020). These two datasets have been widely used for validating satellite-derived LSTs due to their extensive distribution, rigorous quality control, and long-term availability (Guillevic et al., 2018; Martin et al., 2019; Duan et al., 2019).

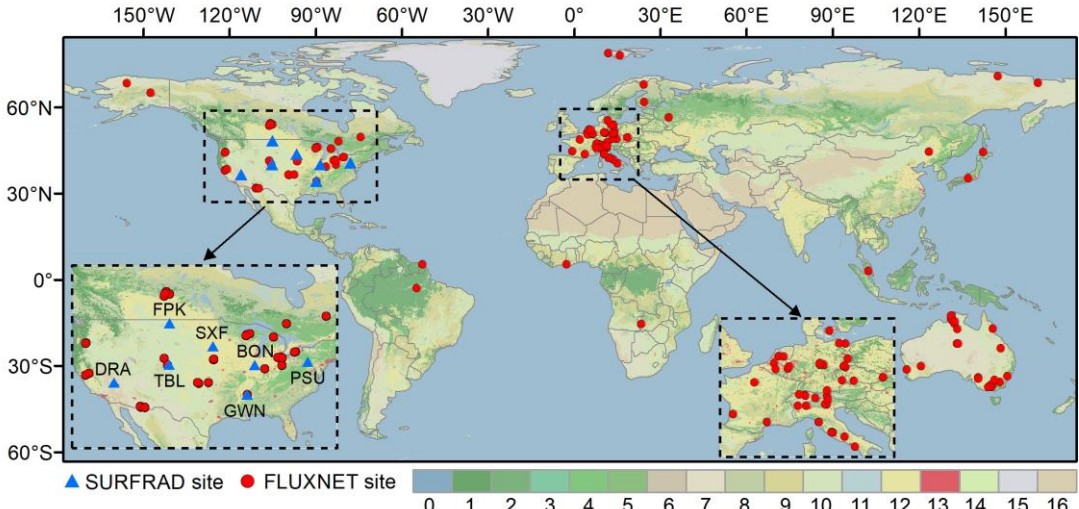

**Fig. 1. Geo-location of the stations used for validation. The red circles and blue triangles represent the locations of the**
**FLUXNET and SURFRAD sites, respectively. The numbers '0' to '16' at the bottommost represent the background**
**land cover type as defined by the International Geosphere–Biosphere Programme (IGBP) (Friedl et al., 2002).**

### 2.3.1 SURFRAD data

We employed observations from the seven SURFRAD sites during the period of 2003 – 2019 (available at
https://www.esrl.noaa.gov/gmd/grad/surfrad/). The seven SURFRAD sites have relatively heterogeneous surfaces and their
land cover types include grassland, cropland, and bare soil. Broadband hemispherical radiances are measured with
pyrgeometers (Eppley Precision Infrared Radiometer) with a wavelength range of 4 – 50 µm. Sensors at each site are
installed at 10-m height with a spatial representativeness of approximately $70 \times 70$ m$^2$ (Guillevic et al., 2014). More detailed
information on these sites is given in Table 1 in Section 4.2. *In situ* LSTs were estimated with the measured upward and
downward longwave radiances with the following formula:

$$
\begin{cases}
T = \sqrt[4]{\dfrac{L^{\uparrow} - (1 - \varepsilon_b) L^{\downarrow}}{\varepsilon_b \sigma}} \\
\varepsilon_b = 0.261 + 0.314 \varepsilon_{31} + 0.411 \varepsilon_{32}
\end{cases}
\tag{1}
$$

where $L^{\uparrow}$ and $L^{\downarrow}$ are the upward and downward longwave radiation, respectively; $\varepsilon_b$ is the broadband emissivity estimated
with the MODIS narrowband emissivities $\varepsilon_{31}$ and $\varepsilon_{32}$ in MODIS Channels 31 and 32, respectively (Liang et al., 2013); and $\sigma$
is the Stefan-Boltzmann constant ($5.67 \times 10^{-8}$ W·m$^{-2}$·K$^{-4}$). To reduce the impacts of outliers on validation, we aggregated
minutely observations into hourly values.



### 2.3.2 FLUXNET data

We further employed the FLUXNET 2015 datasets (available at https://fluxnet.org/data/fluxnet2015-dataset/) to evaluate the GADTC product (Pastorello et al., 2020). The FLUXNET 2015 datasets include more than 200 sites covering multiple ecosystem types across the globe and provide hourly upwelling and downwelling longwave radiation observations of two pyrgeometers (spectral range 3.5 – 50.0 µm) that can be used to retrieve LST (Guillevic et al., 2018). Removing the sites without upwelling longwave radiation observations resulted in a total of 126 sites for the period from 2003 – 2015 (Fig. 1). The *in situ* LSTs were calculated using the same method as for the SURFRAD data.

### 3 Methodology

### 3.1 Generation of global gap-free daily mean LST with the IADTC framework

The OADTC framework consists of two steps to generate $T_{dm}$ (Hong et al., 2021): (1) Reconstruction of instantaneous under-cloud LSTs with an ATC model to ensure the availability of four valid LSTs at the four daily overpass times. (2) Simulation of diurnal LST dynamics using a four-parameter DTC model and estimation of $T_{dm}$. This study improved the OADTC framework by using a more advanced ATC model as well as by optimizing the estimation of $T_{dm}$ with the DTC model. The generation of global gap-free $T_{dm}$ with this improved framework (termed the IADTC framework) includes four steps (Fig. 2): data preprocessing (Section 3.1.1), under-cloud LST reconstruction with an advanced ATC model (Section 3.1.2), linear interpolation of MODIS overpass time (Section 3.1.3), and $T_{dm}$ estimation with a DTC model (Section 3.1.4).



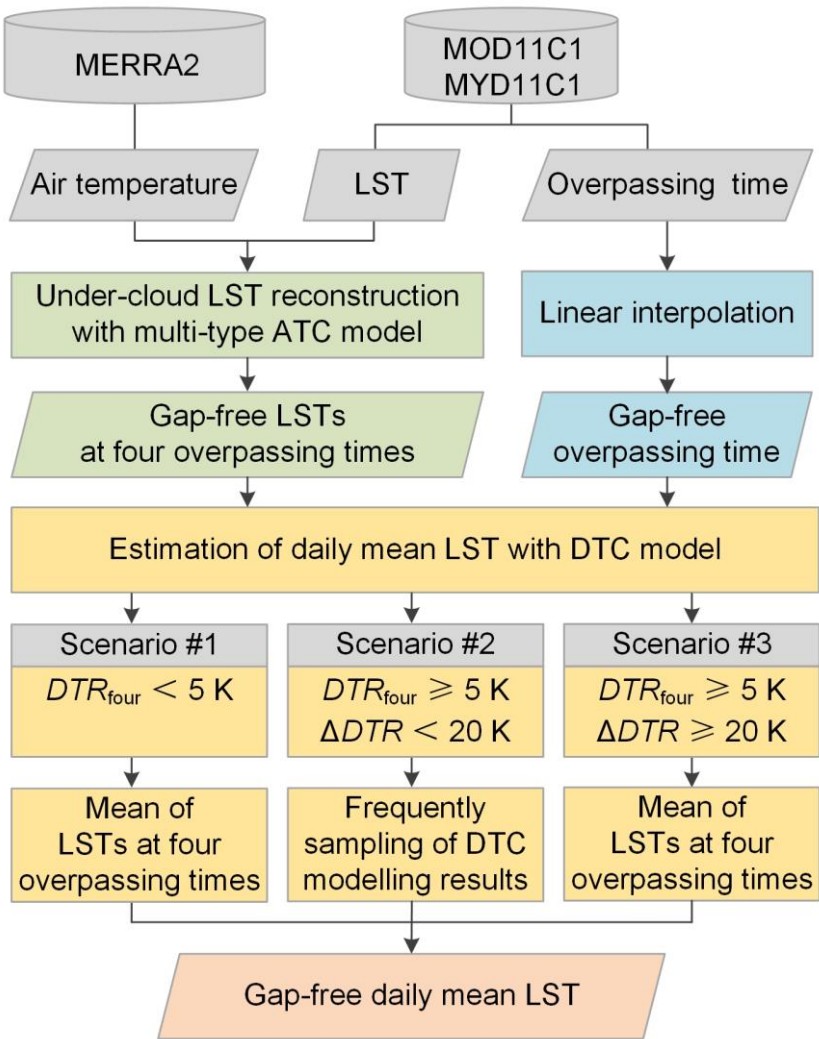

**Fig. 2. Flowchart of the IADTC framework.** $DTR_{four}$ **refers to diurnal temperature range (DTR) calculated as the maximum minus the minimum from the gap-free LSTs at the four overpassing times;** $DTR_{DTC}$ **refers to the DTR calculated from the hourly LSTs modelled with the DTC model.** $\Delta DTR$ **refers to the absolute difference between** $DTR_{four}$ **and** $DTR_{DTC}$**.**

### 3.1.1 Data pre-processing

We generated the global $T_{dm}$ product with a spatial resolution of $0.5 \times 0.5$ degrees rather than a higher resolution (e.g., 1 km) mainly because of the following two aspects. First, our study aims at analyzing the spatial pattern of $\Delta T_{sb}$ and the LST trend at the global scale, i.e., to perform a LST climatology analysis for which a spatial resolution of 0.5 degree is adequate.

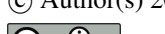



Second, the $T_{dm}$ generation is conducted on a daily and pixel-by-pixel basis on the global scale, which requires a huge amount of computational resources on a higher spatial resolution. Consequently, the MOD11C1 and MYD11C1 products were resampled to a spatial resolution of 0.5 degree; the MERRA2 reanalysis hourly air temperature data were resampled to

daily values with the same resolution.

### 3.1.2 Under-cloud LST reconstruction with multi-type ATC model

The single-type ATC model usually uses a single sinusoidal function to model the intra-annual LST variations driven by solar radiation change and incorporates surface air temperatures to help simulate the LST fluctuations induced by synoptic

conditions (Zou et al., 2018; Liu et al., 2019b). The use of a single sinusoidal function is generally acceptable for mid-latitude regions. However, a single sinusoidal is no longer suitable for low-latitude because there are two solar radiation peaks within a yearly cycle of low-latitude regions; it is also inadequate for high-latitude regions where polar days and nights occur (Bechtel, 2015; Liu et al., 2019b; Xing et al., 2020). Therefore, the use of the single-type ATC model in the OADTC framework is less suitable to generate $T_{dm}$ at the global scale (Fig. 3). To overcome this limitation, the IADTC framework

uses different ATC models (termed the multi-type ATC model) to reconstruct under-cloud LSTs over the low-, mid-, and high-latitude regions, respectively.

Data

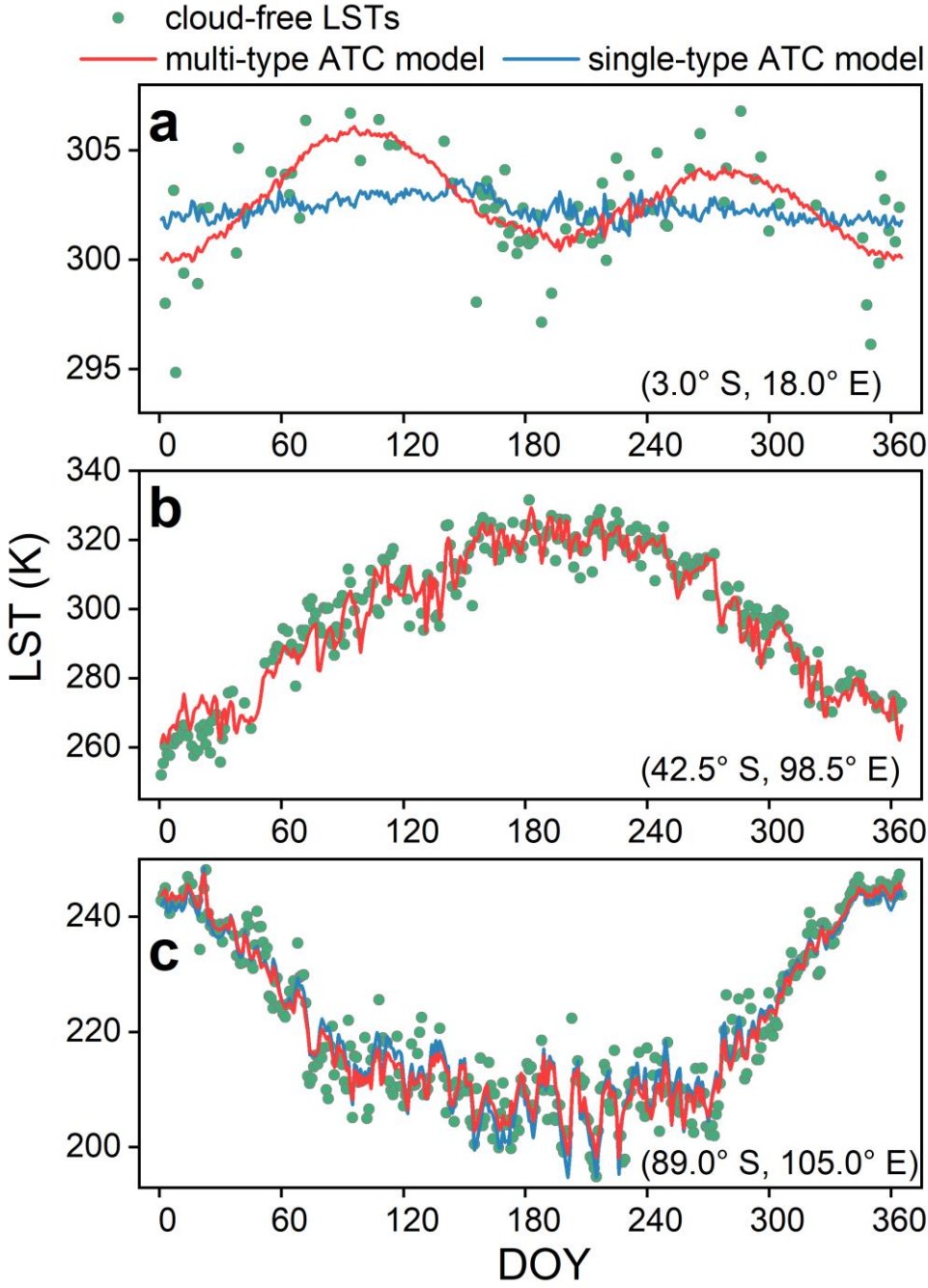

**Fig. 3. Comparison of reconstructing under-cloud LSTs with multi-type and single-type ATC models at different**
**latitudes. (a), (b), and (c) show three examples of ATC modelling at low-latitudes, mid-latitudes, and high-latitudes**
**for cloud-free Terra-day LST in 2019. The green circles, blue lines, and red lines denote the cloud-free observations**



**and LSTs simulated by the single- and multi-type ATC models, respectively. Note that for (b) the results of the single-and multi-type ATC models are identical.**

(1) *Low-latitude regions (23.5° N – 23.5° S)*

The solar radiation possesses two peaks within a yearly cycle over low-latitude regions (Fig. 3a). We therefore employed two sinusoidal functions (termed as the double-sinusoidal ATC model) to reconstruct the daily LST dynamics within an annual cycle (Liu et al., 2019b; Xing et al., 2020). Its formula is as follows:

$$\begin{cases} T_{\text{ATCM}}(d) = T_0 + A_1 \sin\left(\frac{2\pi d}{N} + \theta_1\right) + A_2 \sin\left(\frac{4\pi d}{N} + \theta_2\right) + k \cdot \Delta T_{\text{air}}(d) \\ \Delta T_{\text{air}}(d) = T_{\text{air}}(d) - T_{\text{ATCO}}(d) \\ T_{\text{ATCO}}(d) = T_0^{'} + A_1^{'} \sin\left(\frac{2\pi d}{N} + \theta_1^{'}\right) + A_2^{'} \sin\left(\frac{4\pi d}{N} + \theta_2^{'}\right) \end{cases} \tag{2}$$

where $T_{\text{ATCM}}(d)$ denotes the daily LST variations simulated with the multi-type ATC model; $d$ and $N$ are the day of year
(DOY) and number of days in a year, respectively; $\Delta T_{\text{air}}(d)$ is the difference between the daily SATs (i.e., $T_{\text{air}}(d)$, obtained from MERRA2 reanalysis data) and the modelled air temperatures with the original ATC model ($T_{\text{ATCO}}(d)$); and $T_0$, $A_1$, $\theta_1$, $A_2$, $\theta_2$, and $k$ are six parameters that need to be solved with the cloud-free daily LSTs and SATs, usually through the least-square method.

(2) *Mid-latitude regions (23.5° N/S – 66.5° N/S)*

The solar radiation peaks once in summer during an annual cycle. We therefore employed the single-sinusoidal ATC model (Zou et al., 2018) to reconstruct the daily LST dynamics (Fig. 3b), with its formula as follows:

$$\begin{cases} T_{\text{ATCM}}(d) = T_0 + A \sin\left(\frac{2\pi d}{N} + \theta\right) + k \cdot \Delta T_{\text{air}}(d) \\ \Delta T_{\text{air}}(d) = T_{\text{air}}(d) - T_{\text{ATCO}}(d) \\ T_{\text{ATCO}}(d) = T_0^{'} + A^{'} \sin\left(\frac{2\pi d}{N} + \theta^{'}\right) \end{cases} \tag{3}$$

where $T_0$, $A$, $\theta$, and $k$ are the four parameters. The parameter meanings and their computation are the same as those of the double-sinusoidal ATC model.

(3) *High-latitude regions (66.5° N/S – 90° N/S)*

The polar day/night phenomena occur over high-latitude regions and the duration increases with the latitude. Theoretically, over these regions, the multi-sinusoidal ATC model should be the best choice. However, the number of cloud-free MODIS observations is limited, and additional model complexity can lead to over-fitting and weaken the generalization ability of the ATC model (Liu et al., 2019b). To balance model accuracy and generalization ability, the double-sinusoidal ATC model was
selected for high-latitude regions (see Fig. 3c and Eq. 2).

### 3.1.3 Interpolation of overpassing times

The under-cloud LST reconstruction with the ATC model ensures that there are four valid LSTs within a diurnal cycle. However, there are still missing values for the corresponding four overpassing times. We used linear interpolation to reconstruct the missing overpassing times based on the valid overpassing times on the two adjacent days with valid values.





For example, if the overpassing times from Jul 10$^{th}$ to Jul 20$^{th}$ for Aqua day are missing, the linear interpolation was used to fill the missing values during this period using the valid values on the two adjacent days with valid values (i.e., Jul 9$^{th}$ and Jul 21$^{st}$). The uncertainties of linear interpolation are expected to be within the range associated with local overpassing time fluctuations. For the low- and mid-latitude regions where the overpassing time fluctuations are relatively small (less than 1.5 hours), the uncertainties using linear interpolation are relatively minor. However, for the high-latitude regions where the

overpassing times fluctuate significantly (Fig. A1), linear interpolation holds a larger error and might affect the estimation of $T_{dm}$. More discussions in terms of the uncertainties of the linear interpolation are provided in Section 5.2.

### 3.1.4 Estimation of daily mean LST with DTC model

The under-cloud LST reconstruction (Section 3.1.2) and linear interpolation of overpassing time (Section 3.1.3) ensure that there are four valid LSTs and the associated overpassing times per day. These provide the foundation for estimating $T_{dm}$ with

a four-parameter DTC model. This study selected the four-parameter GOT09-dT-τ model, which has been shown to have the highest accuracy among a variety of four-parameter DTC models (Hong et al., 2018). Further details related to the formulae and the associated parameters of the GOT09-dT-τ model are provided in Göttsche and Olesen (2009) and Hong et al. (2018).

For the generation of global products, the GOT09-dT-τ model can face the issues of no-solution or extreme-solution, under which the estimated $T_{dm}$ can be significantly biased due to the reduced capability to model LST around sunrise (Hu et

al., 2020) (Fig. 4a). The failed simulations can be associated with the following two reasons: (1) there are four daily MODIS LSTs per daily cycle but no observation around sunrise (Hong et al., 2018); (2) the DTC model is subject to the clear-sky hypothesis (Göttsche and Olesen, 2009). Therefore, to avoid outliers caused by failed simulations, under certain conditions, $T_{dm}$ was estimated directly by averaging the four LSTs per daily cycle. We introduced two criteria to determine whether to use the DTC model for estimating $T_{dm}$ or not (Fig. 2, Scenario #1 to #3).

The first criterion is based on the diurnal temperature range (DTR), which was calculated as the maximum minus the minimum LSTs within a diurnal cycle. Specifically, the DTR calculated by four LSTs within the diurnal cycle (termed $DTR_{four}$) was used (Fig. 2). For relatively small $DTR_{four}$, e.g., on overcast days with heavy clouds or on days with low incoming solar radiation (e.g., polar nights), $T_{dm}$ can be directly estimated as the mean of the four daily LSTs per daily cycle. In this case, the DTC model would be unnecessary. We empirically set the $DTR_{four}$ threshold as 5.0 K (see Section 5.1 for

detailed discussions). In other words, when the $DTR_{four}$ is less than 5.0 K, (see Scenario #1 in Fig. 2), $T_{dm}$ estimated with the IADTC framework (termed $T_{dm\_IADTC}$) was obtained by averaging the four LSTs within a diurnal cycle (termed $T_{dm\_ATC\_four}$). Here these four daily LSTs can consist of both cloud-free observations ($T_{in\_cloud\_free}$, the green circles in Fig. 4) and under-cloud LSTs reconstructed by the ATC model ($T_{in\_ATC}$, the blue triangles in Fig. 4).

When $DTR_{four}$ is greater than 5.0 K, the DTC model would be used to simulate the diurnal LST dynamics. However, for

the global generation of $T_{dm}$, the simulation can still fail for cases with complicated diurnal LST dynamics (Fig. 4a). To avoid this issue, we introduced the second criterion to determine whether to use the DTC model or not. This was done by comparing the $DTR_{four}$ and the DTR calculated by the DTC model (termed $DTR_{DTC}$). This comparison can be used to identify

the failed simulations of the DTC model because the $DTR_{DTC}$ would be abnormal once the LSTs modelled by the DTC model are significantly underestimated around sunrise. Therefore, we employed the absolute difference between $DTR_{DTC}$ and

$DTR_{four}$ (termed as $\Delta DTR$) as the second threshold to further determine whether to use the DTC model or not. This study empirically set the $\Delta DTR$ threshold as 20.0 K. More discussions on this is provided in Section 5.1.

In the practical generation of $T_{dm}$, when $DTR_{four} \geq 5.0$ K and $\Delta DTR < 20.0$ K (Scenario #2 in Fig. 2), the DTC modelling results ($T_{in\_ATC\_DTC}$, see the blue line in Fig. 4b) are satisfactory and were then used to estimate $T_{dm}$. The $T_{dm\_IADTC}$ was then calculated as the average of instantaneous hourly LSTs ($T_{in\_ATC\_DTC}$). When $DTR_{four} \geq 5.0$ K and $\Delta DTR \geq 20.0$ K (Scenario

#3 in Fig. 2), the DTC model may fail (Fig. 4a) as the $T_{dm}$ estimate based on the DTC modelling (i.e., $T_{dm\_ATC\_DTC}$) is considerably lower than the true $T_{dm}$. In this case, the error of $T_{dm\_ATC\_DTC}$ can be even larger than that of $T_{dm}$ estimated as the average of the four LSTs within the day (i.e., $T_{dm\_ATC\_four}$; refer to Fig. 11 in Section 5.1). Therefore, in this case $T_{dm\_IADTC}$ was directly calculated as $T_{dm\_ATC\_four}$. In summary, for Scenarios #1 and #3, $T_{dm\_IADTC}$ was calculated as $T_{dm\_ATC\_four}$, while it was calculated as $T_{dm\_ATC\_DTC}$ for Scenario #2.

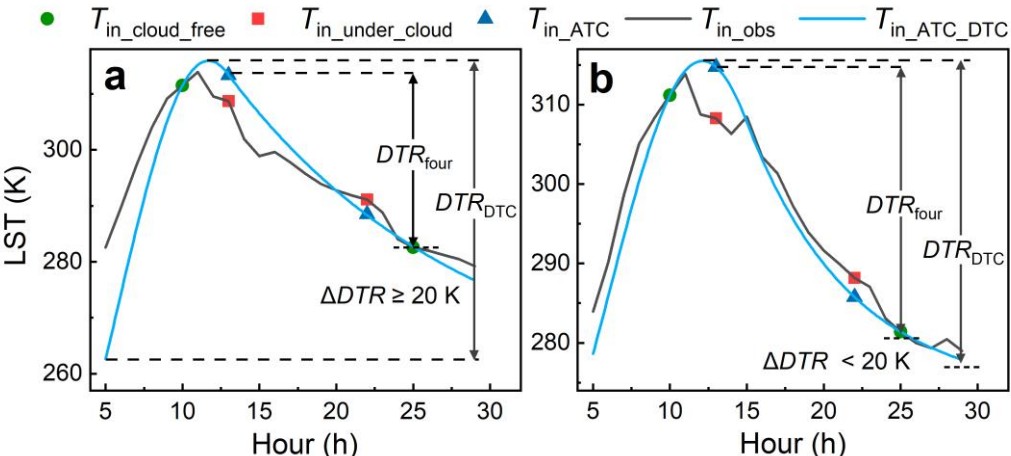

**Fig. 4. Estimation of $T_{dm}$ under different conditions. (a) displays an example of estimating $T_{dm}$ by averaging $T_{in\_cloud\_free}$ and $T_{in\_ATC}$ when $\Delta DTR$ is equal or greater than 20.0 K (i.e., Scenario #3); (b) displays an example of estimating $T_{dm}$ based on the DTC modelling results (i.e., Scenario #2). The green circles, red rectangles, and blue triangles denote the instantaneous cloud-free LST observations, under-cloud LST observations, and under-cloud**

**LSTs reconstructed by the ATC model, respectively. The black lines denote the *in situ* LST observations while the blue lines show the DTC-modelled values based on the cloud-free LST observations and ATC-modelled under-cloud LSTs. Noting that hours larger than 24 along the x-axis correspond to the next day.**

### 3.2 Validations

The GADTC products were validated from the following two aspects: (1) validating the IADTC framework indirectly with

single-source *in situ* measurements at the site level; and (2) validating the GADTC products directly by comparing with *in*

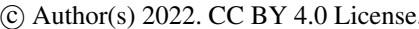



*situ* measurements. These two aspects complement each other and allow to assess the applicability of IADTC framework and the accuracy of the generated GADTC products. The direct comparison of the GADTC product with *in situ* measurements (SURFRAD and FLUXNET measurements for this study) provide information on the accuracy of the IADTC framework especially over homogeneous areas (Guillevic et al., 2018). However, such direct validations can be affected by uncertainties

beyond the IADTC framework, e.g., a mismatch of spatial scale between satellite and *in situ* measurements, different observation angles, and uncertainties from the LST retrieval algorithm (Ermida et al., 2014; Guillevic et al., 2014; Li et al., 2014). Therefore, direct comparisons may not fully reflect the true accuracy of the IADTC framework. To address this issue and assess the applicability of IADTC framework, we validated the IADTC framework indirectly by driving it with *in situ* measurement and then using hourly measurements for validation. This strategy avoids the mismatch issue of multi-source

data and can, therefore, better reflect the accuracy of the IADTC framework (Hong et al., 2021).

### 3.2.1 Validation of the IADTC framework with in situ measurements

The IADTC framework was validated with *in situ* hourly measurements obtained exclusively from SURFRAD and FLUXNET data. During this validation process, the MERRA2 air temperature at the corresponding station location, instead of the air temperature from *in situ* measurements, were used to drive ATC model, which is identical to the actual generation

of the GADTC products.

The approach used the cloud-free *in situ* measurements at each MODIS overpassing time and MERRA2 air temperatures to drive the ATC model, and the corresponding under-cloud *in situ* measurements ($T_{\text{in\_under\_cloud}}$, red rectangles in Fig. 4) to evaluate the accuracy of the under-cloud LSTs reconstructed by the ATC model ($T_{\text{in\_ATC}}$). The accuracy of the $T_{\text{dm}}$ estimated with the IADTC framework ($T_{\text{dm\_IADTC}}$) was evaluated against 'true' $T_{\text{dm}}$ (termed $T_{\text{dm\_true}}$), i.e., the average of

the hourly *in situ* measurements ($T_{\text{in\_obs}}$, gray line in Fig. 4). We also provided the sampling bias ($\Delta T_{\text{sb}}$) of the traditional method based on cloud-free observations (i.e., the average of $T_{\text{in\_cloud\_free}}$), which here is termed $T_{\text{dm\_cloud\_free}}$. Therefore, the accuracy improvements of $T_{\text{dm\_IADTC}}$ compared to $T_{\text{dm\_cloud\_free}}$ are reflected in the corresponding reduction of $\Delta T_{\text{sb}}$. We further provide $T_{\text{dm}}$ estimated with the OADTC framework (termed $T_{\text{dm\_OADTC}}$) to illustrate the improvement achieved by the IADTC framework.

### 3.2.2 Validation of the GADTC product directly with in situ measurements

After matching the geolocation and observation time, we directly compared the GADTC product with *in situ* $T_{\text{dm}}$ measurements from SURFRAD and FLUXNET. Note that outliers in the *in situ* measurements were removed before performing the accuracy evaluation; here outliers are defined as LST values deviating by more than 3σ (three standard deviations) from the mean (Zhang et al., 2020).





### 3.3 Analysis of the GADTC product

We analyzed the difference in LST values and trends between $T_{dm\_cloud\_free}$ (the daily mean LST estimated by the traditional method) and the GADTC products. For the difference in LST values, we present the global spatial distribution of $\Delta T_{sb}$ by using the GADTC product as the reference (see Section 4.3). For the difference in LST trends, the seasonal Mann–Kendall test and Theil-sen slope were used to diagnose the warming/cooling trend of LST and describe its slope, respectively (see Section 4.4). The seasonal Mann–Kendall test is a non-parametric test suitable to detect LST warming/cooling trends and to quantify the associated significance level in LST time series (Hirsch et al., 1982; Hussain and Mahmud, 2019), while the Theil-sen slope reduces the impact of outliers on LST trend analysis (Sen, 1968; Theil, 1950). We conducted a seasonal Mann–Kendall test for both the $T_{dm\_cloud\_free}$ and the GADTC product and compared their Theil-sen slopes in describing global LST trends.

## 4. Results

### 4.1 Validation of the IADTC framework with *in situ* measurements

The validations using the SURFRAD measurements show that the MAE and bias of the ATC model for the day are 4.7 K and 4.0 K, respectively, while those for the night are 3.6 K and −1.6 K, respectively (Fig. 5a & Fig. 5c). Although the results for the ATC model are less satisfactory, the $T_{dm}$ accuracies estimated with the IADTC framework is generally acceptable: the MAEs of $T_{dm\_IADTC}$ at the daily and monthly scales are 1.4 K and 0.6 K, respectively and the corresponding biases are both −0.2 K (Fig. 6). By contrast, the MAEs of the $T_{dm\_cloud\_free}$ are 4.1 K and 2.5 K at the daily and monthly scales, respectively, i.e., they indicate a significantly lower accuracy.

Compared with the OADTC framework, the IADTC framework improves the MAE of estimated $T_{dm}$ by around 0.45 K (from 2.80 K to 2.35 K, see Fig. B1a) under Scenarios #1 and #3. The accuracy improvement results mainly from two aspects: (1) the IADTC framework reduces the systematic negative bias caused by cases for which the DTC-modelled LSTs are significantly underestimated around sunrise; and (2) the outliers due to failed DTC simulations are avoided. The overall accuracies for all three scenarios show that the IADTC framework improves the bias from −0.38 K to −0.18 K, while the MAE improvement is relatively small. The relatively slight increase in the overall accuracy is attributed to the relatively small proportion of Scenarios #1 and #3 (around 5%).



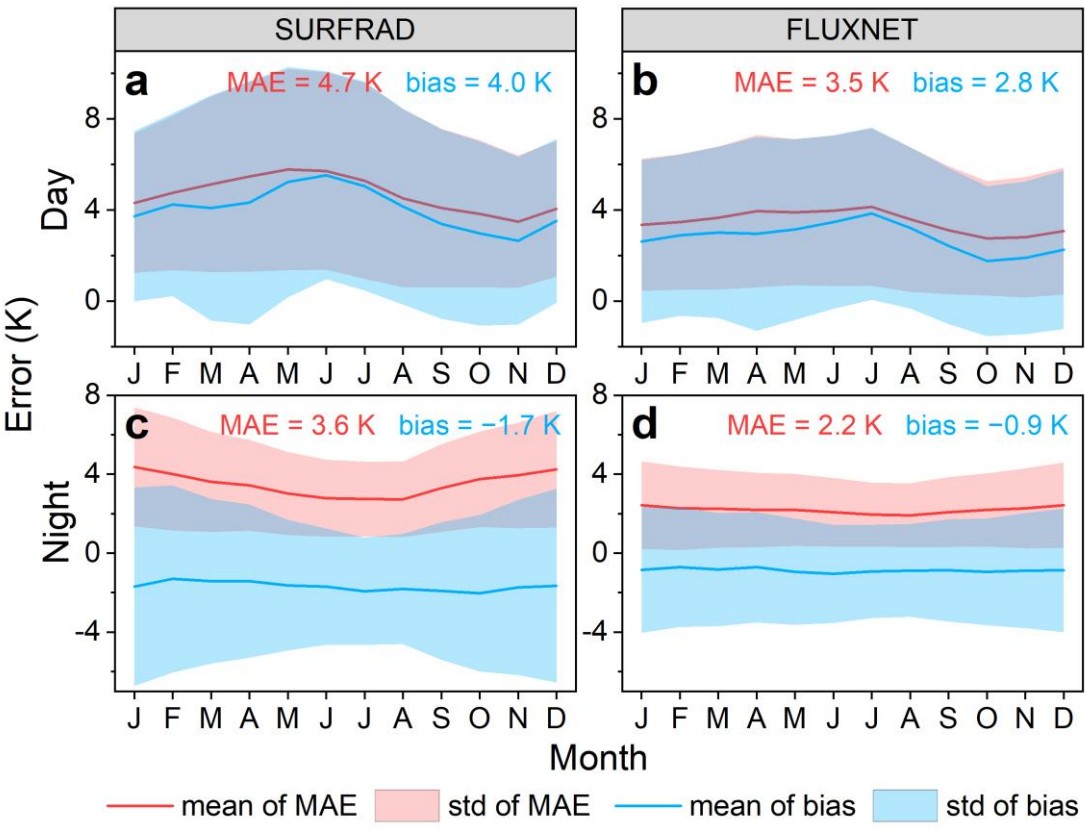

325

**Fig. 5. Validations of reconstructed under-cloud LSTs at Aqua and Terra day and night overpass times based exclusively on *in situ* data. The under-cloud LSTs were reconstructed with the ATC model. (a) and (b) show monthly mean errors obtained for daytime overpasses (including Aqua-day and Terra-day) for SURFRAD and FLUXNET data, respectively; (c) and (d) show the same for the nighttime overpasses (including Aqua-night and Terra-night).**



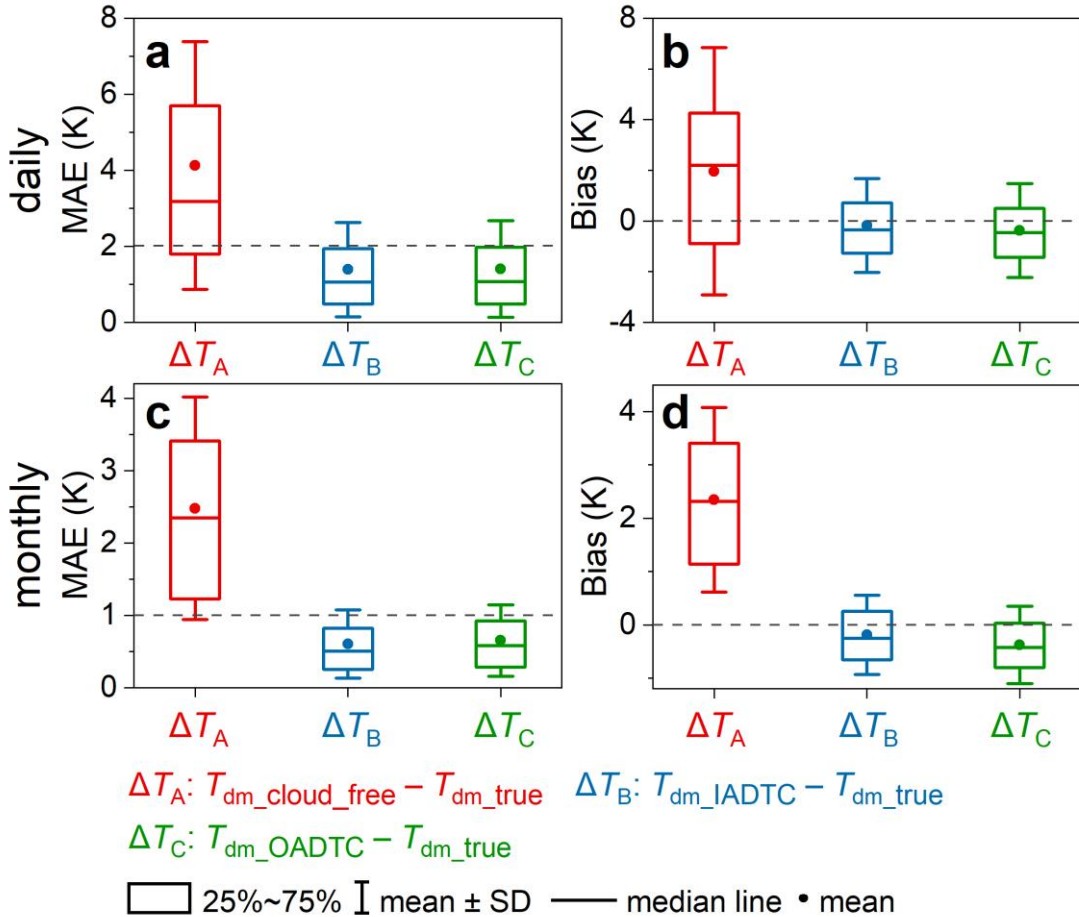

**Fig. 6. Validations of daily mean LST ($T_{dm}$) estimation with SURFRAD data. Boxplots show the errors for the traditional $T_{dm}$ estimation method ($T_{dm\_cloud\_free}$), the IADTC framework ($T_{dm\_ATC\_DTC}$), and the OADTC framework ($T_{dm\_OADTC}$). (a) and (b) display the MAE and bias at the daily scale, respectively; (c) and (d) display the MAE and bias at the monthly scale, respectively.**

The validations using the FLUXNET data are similar to those with the SURFRAD data: (1) the IADTC framework significantly reduces the $\Delta T_{sb}$ of $T_{dm\_cloud\_free}$; (2) the MAEs of $T_{dm\_IADTC}$ are 1.1 K and 0.5 K at the daily and monthly scales, respectively; and (3) the biases are both close to zero (Fig. 7). The validations again indicate that the under-cloud LSTs reconstructed by the ATC model are systematically positive during the day (the MAE and bias are 3.5 K and 2.8 K, respectively) and systematically negative during the night (the MAE and bias are 2.2 K and −0.9 K, respectively) (Fig. 5b & Fig. 5d). Compared with the OADTC framework, in Scenarios #1 and #3 (the proportion is 16%) under which the accuracies are considerably improved, IADTC framework improved the MAE of the estimated $T_{dm}$ by around 0.78 K (from 1.95 K to 1.17 K, refer to Fig. B1b). However, for all the three scenarios, the overall MAE and bias improvements of the IADTC framework are around 0.15 K and 0.30 K, respectively (Fig. 7).



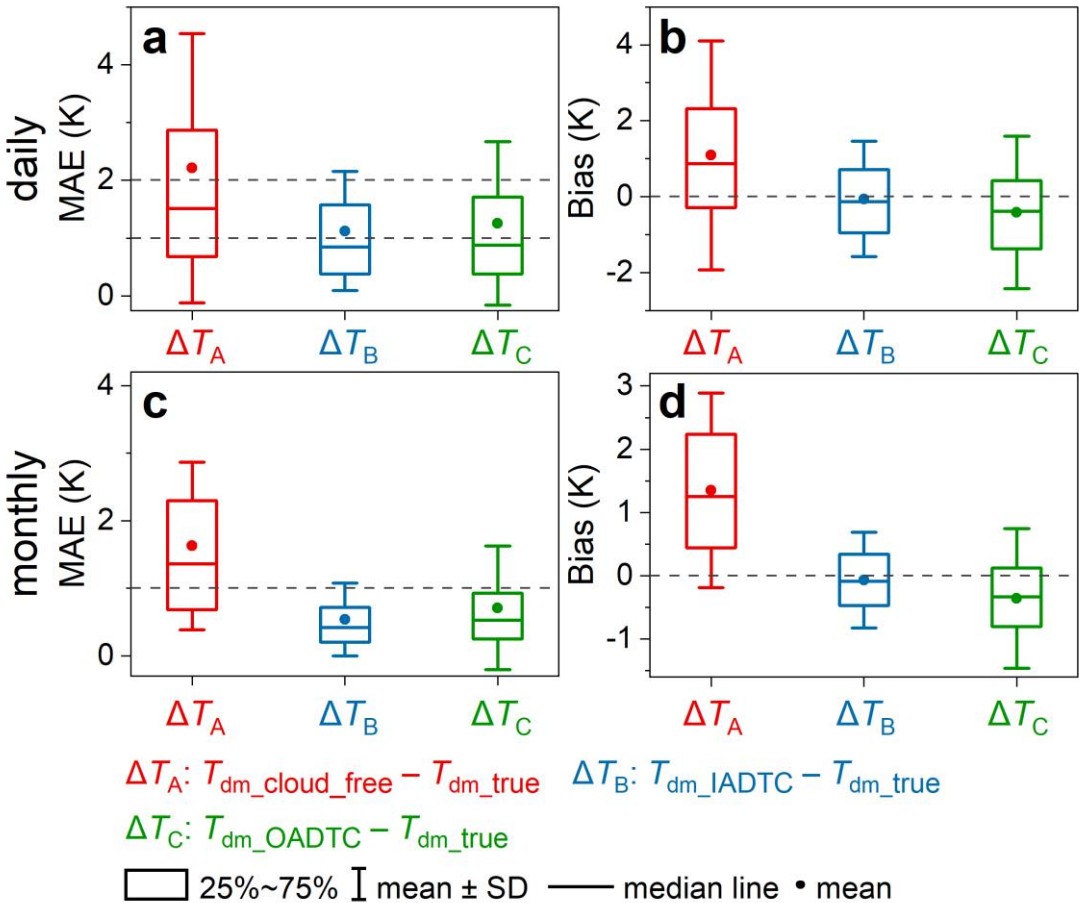

Fig. 7. The same as Fig. 6, but for the FLUXNET data.

## 4.2 Evaluation of the GADTC product with *in situ* measurements

The comparison between the GADTC products and *in situ* measurements (SURFRAD and FLUXNET datasets) shows that the MAEs of the GADTC products are 3.0 K and 2.6 K at the daily and monthly scales, respectively, and the mean bias on both scale is −1.5 K (Fig. 8). The MAE and bias are larger than those of the IADTC framework at site level (Fig. 6). This is thought to be due to inconsistencies between MODIS cloud-free observations and *in situ* measurements, i.e., errors of MODIS cloud-free observations propagating into the GADTC products. The mismatch in spatial resolution between the GADTC products and *in situ* measurements can also lead to lower accuracies.

The validation with the SURFRAD measurements show that the MAE of the GADTC products is 2.2 K and 1.6 K at the daily and monthly scales, respectively and the bias is around −1.6 K at both scales (Fig. 8a & Fig. 8c). These accuracies of daily mean LST are generally on par with those of instantaneous LSTs in studies comparing instantaneous MODIS cloud-free observations and SURFRAD measurements (Duan et al., 2019; Martin et al., 2019). Across the different SURFRAD



sites, the MAEs of the GADTC products are relatively similar (around 2.2 K; see Table 1), including those for the DRA site (the background land cover type is bare soil), at which significant biases have been detected between instantaneous MODIS cloud-free observations and SURFRAD measurements (Duan et al., 2019; Ermida et al., 2020).


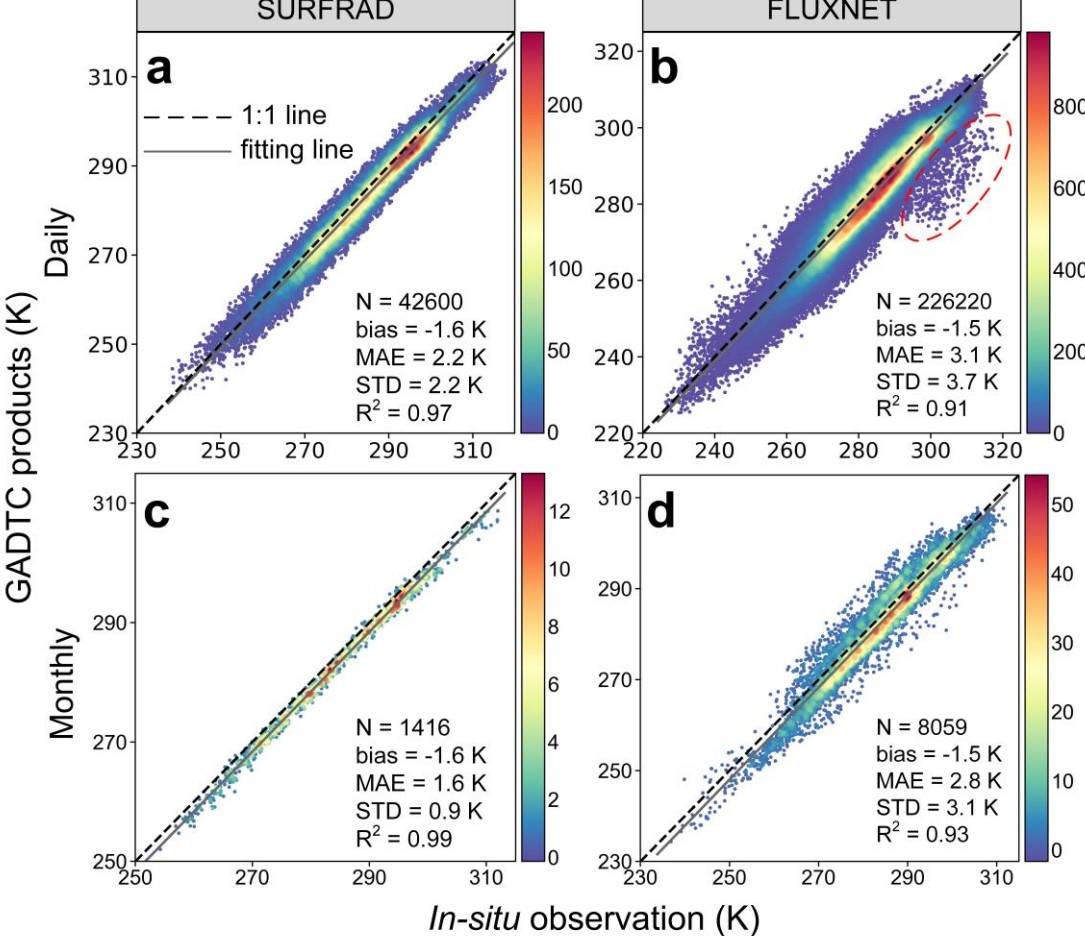

**Fig. 8. GADTC products versus *in situ* observations. (a) and (b) compare the daily mean LST over the SURFRAD and FLUXNET sites, respectively; and (c) and (d) show the corresponding results for monthly mean LST. The biases are calculated by the GADTC products minus the *in situ* measurements. The red ellipse in (b) highlights cases with**

**notably large errors.**

**Table 1. Validation results obtained over the seven SURFRAD sites.**

| Site ID | Lat./Long. | IGBP | N* | Bias (K) | MAE (K) | STD (K) |
|---------|------------|------|-----|----------|---------|---------|
| BON | 40.05°/−88.37° | CRO | 6153 | −1.20 | 1.97 | 2.12 |
| TBL | 40.13°/−105.24° | GRA | 6124 | −1.37 | 2.30 | 2.54 |



| | | | | | | |
|---|---|---|---|---|---|---|
| DRA | 36.62°/−116.02° | BSV | 6102 | −2.04 | 2.26 | 1.74 |
| FPK | 48.31°/−105.10° | GRA | 6157 | −1.78 | 2.54 | 2.63 |
| GWN | 34.25°/−89.87° | WSA | 6144 | −1.83 | 2.25 | 1.98 |
| PSU | 40.72°/−77.93° | CRO | 6134 | −1.30 | 1.85 | 1.82 |
| SXF | 43.73°/−96.62° | CRO | 5786 | −1.39 | 2.06 | 2.13 |

*: $N$ denotes the number of days used for validation.

The direct comparison between the GADTC products and FLUXNET measurements shows that the MAEs are 3.1 K and 2.8 K at the daily and monthly scales, respectively; and the bias at these two time scales is −1.5 K (Fig. 8b & Fig. 8d).
Compared with the validations over the SURFRAD sites, the accuracies over the FLUXNET sites decrease slightly and the standard deviations increases. The relatively larger errors at several FLUXNET sites (e.g., AU-Wac, SJ-Adv, and CH-Fru sites, with MAEs larger than 8.0 K; refer to the red ellipse in Fig. 8b) partly account for the lower accuracy. The relatively large errors at these sites might be related to the erroneous *in situ* measurements as well as the high spatial heterogeneity around these sites. However, the accuracies at most FLUXNET sites are acceptable.

The validations over the FLUXNET sites show that the MAEs vary from 2.6 to 4.8 K and depend on land cover type. Relatively lower accuracies of the GADTC products (MAE larger than 3.5 K) are observed over IGBP land cover types OSH (Open Shrublands) and SNO (Snow and Ice) (Table 2). This may be related to unusually large measurement errors and the relatively high spatial heterogeneity at some sites as well as the limited number of sites representing a particular land cover type. For example, the accuracy assessment over the SNO land cover type is performed with a single site and there are only 380 three sites of the OSH land cover type (e.g., the RU-Cok with MAE as large as 4.6 K).

**Table 2. Validation results for the GADTC products stratified by IGBP land cover type of the FLUXNET sites.**

| IGBP | Site number | $N$* | Bias (K) | MAE (K) | STD (K) |
|---|---|---|---|---|---|
| MF | 5 | 7564 | −1.95 | 2.62 | 2.61 |
| EBF | 11 | 29588 | −1.71 | 2.75 | 2.87 |
| WET | 15 | 14556 | −0.66 | 2.76 | 4.17 |
| DBF | 19 | 32594 | −1.78 | 2.89 | 3.08 |
| SAV | 5 | 10355 | −2.65 | 3.16 | 2.79 |
| CRO | 14 | 14387 | −1.59 | 3.26 | 3.78 |
| GRA | 23 | 45257 | −1.62 | 3.32 | 3.90 |
| ENF | 25 | 58616 | −0.81 | 3.38 | 4.10 |
| WSA | 5 | 7810 | −2.33 | 3.44 | 3.32 |
| OSH | 3 | 5090 | −3.34 | 3.62 | 2.75 |
| SNO | 1 | 403 | −3.39 | 4.80 | 4.84 |

*: $N$ denotes the number of days used for validation.





### 4.3 Analysis of the GADTC product

The validations based exclusively on *in situ* LST measurements (Fig. 6) show that the IADTC framework can reduce the

sampling bias ($\Delta T_\mathrm{sb}$, i.e., $T_\mathrm{dm\_cloud\_free} - T_\mathrm{dm\_true}$) significantly, especially at the monthly scale. $\Delta T_\mathrm{sb}$ directly affects the value of $T_\mathrm{dm}$ and may further influence the LST trend. Therefore, based on the GADTC products, we analyzed the global distribution of $\Delta T_\mathrm{sb}$ (calculated by $T_\mathrm{dm\_cloud\_free} - T_\mathrm{dm\_IADTC}$) at the monthly scale (Section 4.3.1) and compared the LST trend differences between monthly averaged $T_\mathrm{dm\_cloud\_free}$ and $T_\mathrm{dm\_IADTC}$ to study the impact of $\Delta T_\mathrm{sb}$ on LST trends (Section 4.3.2).

### 4.3.1 Global distribution of the sampling bias $\Delta T_\mathrm{sb}$

The global distribution of the averaged $\Delta T_\mathrm{sb}$ from 2003 to 2019 shows that the global mean $\Delta T_\mathrm{sb}$ is 1.8 K (Fig. 9). At low- and mid-latitude regions, $\Delta T_\mathrm{sb}$ is generally around 2.0 K, yet it can exceed 4.0 K in some regions, e.g., deserts. At high-latitude regions, $\Delta T_\mathrm{sb}$ is close to or slightly less than zero. $\Delta T_\mathrm{sb}$ also varies with month or season (Fig. C1). For example, the average $\Delta T_\mathrm{sb}$ for September-October-November (2.0 K) is larger than that for December-January-February (1.5 K). We further observe that $\Delta T_\mathrm{sb}$ is sensitive to land cover type and that DTR appears to be strongly linked to $\Delta T_\mathrm{sb}$. For instance,

regions with a large DTR (e.g., deserts or bare soils) usually have a greater $\Delta T_\mathrm{sb}$ (Hong et al., 2021; Jin and Dickinson, 2010).

Apart from the DTR, in high-latitude regions, $\Delta T_\mathrm{sb}$ can also be affected by the drift of MODIS overpassing time. The DTR is relatively small in high-latitude regions where the angle of the incident solar radiation is low and the LST observations across a diurnal cycle are often already close to the true $T_\mathrm{dm}$, leading to a relatively small $\Delta T_\mathrm{sb}$. The time drift at

high-latitude regions can also contribute to the relatively small $\Delta T_\mathrm{sb}$. At low- and mid-latitude regions, MODIS samples the surface near 10:30, 13:30, 22:30, and 01:30 (local solar time) (Fig. A1): the systematic positive $\Delta T_\mathrm{sb}$ is then mostly due to the under-sampling of the nighttime cooling until the sunrise of the next day (Hong et al., 2021). At high-latitude regions, the time drift effect allows MODIS observations at other than these four times and alleviates the under-sampling of nighttime cooling, thereby reducing $\Delta T_\mathrm{sb}$.



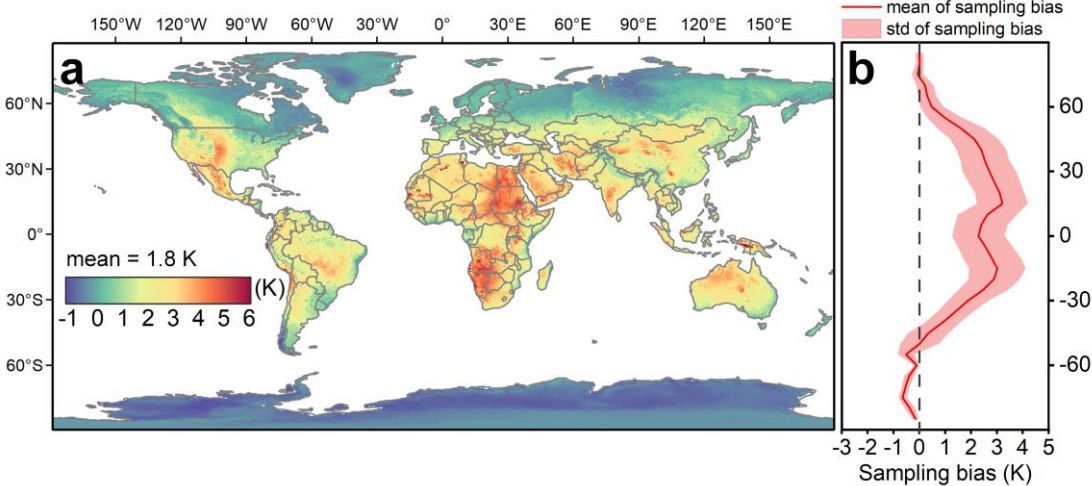


**Fig. 9. Average sampling bias $\Delta T_{sb}$ from 2003 to 2019. (a) global spatial distribution of $\Delta T_{sb}$; and (b) average results for 5-degree intervals along the longitude.**

### 4.3.2 Analysis of global LST trends from 2003 to 2019

The LST trends determined for $T_{dm\_cloud\_free}$ and $T_{dm\_IADTC}$ shows similar global patterns, i.e., both can show comparable

warming/cooling trends (Fig. 10a & Fig. 10b). For example, they both display overall increasing LST trend over the globe as well as an accelerated surface warming trend over the Arctic and Europe (Fig. 10), which is consistent with most previous studies (Mao et al., 2017; Sobrino et al., 2020a; Sobrino et al., 2020b).

However, the slopes of the LST trends are significantly different between $T_{dm\_cloud\_free}$ and $T_{dm\_IADTC}$ with a MAE of 0.012 K/year (Fig. 10e). This difference is related to the different information contained within the $\Delta T_{sb}$ and reflects the

cloud percentage and cloud duration among different months. When taking the slope of $T_{dm\_IADTC}$ as reference, the slope of $T_{dm\_cloud\_free}$ underestimates the global LST warming rate by 0.004 K/year. With the original MODIS LST observations (i.e., $T_{dm\_cloud\_free}$) as reference, the warming LST trends would be underestimated over South America, Africa, Asia, and Oceania. They would be overestimated over Europe and relatively similar to the trends obtained with $T_{dm\_IADTC}$ over North America and Antarctica.



**Fig. 10. Global LST trends from 2003 to 2019. (a) and (b) display the global LST trends based on $T_{dm\_cloud\_free}$ and their averaged results for 5-degree intervals along the longitude, respectively; and (c) and (d) shows the**



**corresponding results for $T_{dm\_IADTC}$; and (e) and (f) show the global LST trend differences between $T_{dm\_cloud\_free}$ and $T_{dm\_IADTC}$ and their averages for 5-degree intervals along the longitude, respectively.**

**5. Discussion**

**5.1 Empirical determination of the threshold for optimizing the $T_{dm}$ estimation with DTC model**

To determine the threshold for the first criterion (i.e., the threshold for the $DTR_{four}$, see Fig. 2), we analyzed the variations in the error of $T_{dm\_ATC\_four}$ depending on $DTR_{four}$ using SURFRAD and FLUXNET data (Fig. 11a & Fig. 11b). The assessments show that the errors of $T_{dm\_ATC\_four}$ generally increase with $DTR_{four}$. The linear fitting lines show that the error of $T_{dm\_ATC\_four}$

is relatively low when $DTR_{four}$ is small. In other words, the direct average of the four LSTs per daily cycle ($T_{dm\_ATC\_four}$) is a good estimate of $T_{dm}$ when the $DTR_{four}$ is small. Based on the linear fits in Fig. 11a & Fig. 11b, we therefore chose the $DTR_{four}$ threshold of the first criterion to be 5.0 K.

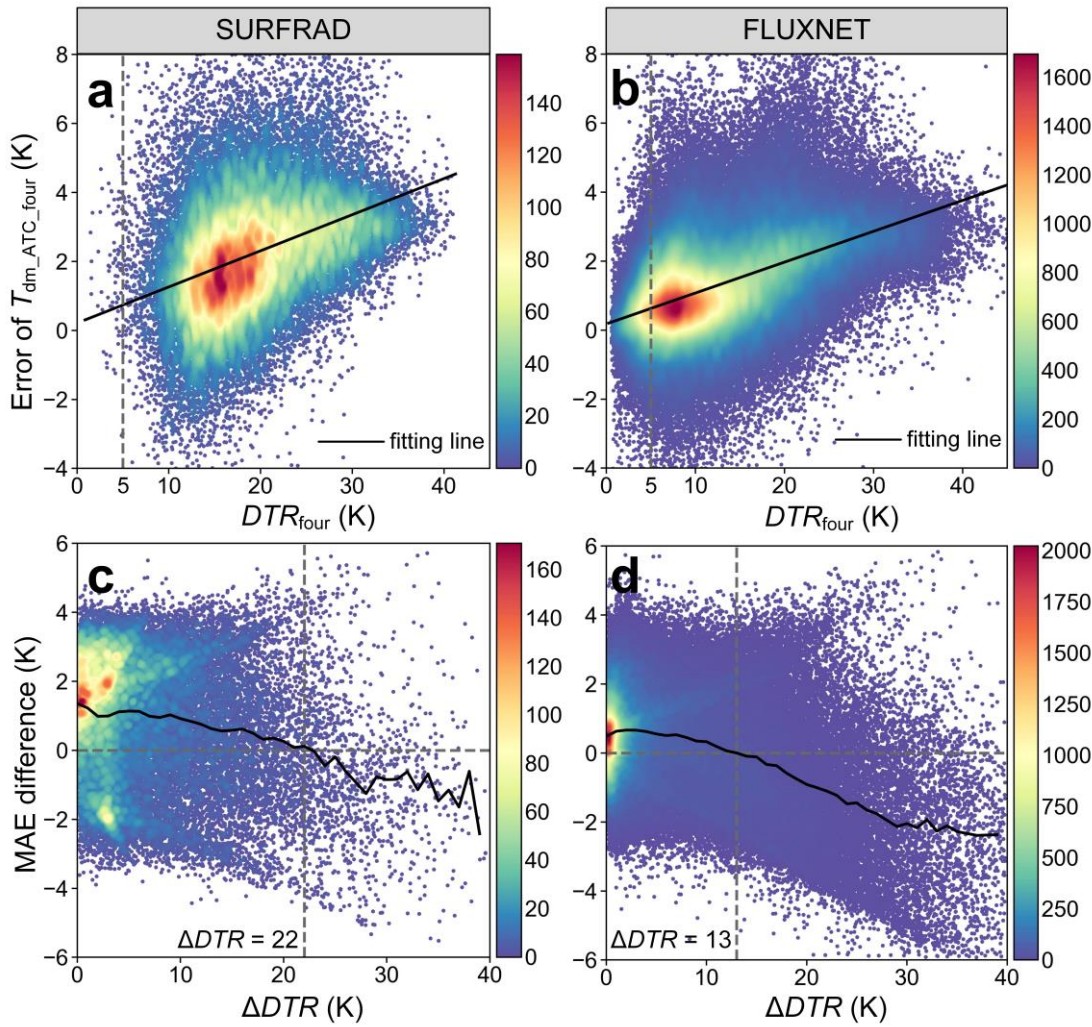

**Fig. 11. Threshold determination for the two criteria in Fig. 2. (a) and (b) display the errors of $T_{dm\_ATC\_four}$**
**($T_{dm\_ATC\_four}$ minus $T_{dm\_true}$) depending on $DTR_{four}$ for SURFRAD and FLUXNET data, respectively; and (c) and (d)**
**display the MAE differences between $T_{dm\_ATC\_four}$ and $T_{dm\_ATC\_DTC}$ (i.e., the MAE of $T_{dm\_ATC\_four}$ minus the MAE of**
**$T_{dm\_ATC\_DTC}$) depending on the $\Delta DTR$ for SURFRAD and FLUXNET data, respectively. The black lines in (c) and (d)**
**denote the averaged MAE difference within every unit along the x-axis.**

The second criterion uses the $\Delta DTR$ to filter cases for which $T_{dm}$ is significantly underestimated. To determine the
optimal threshold for $\Delta DTR$, we analyzed the MAE differences between $T_{dm\_ATC\_four}$ and $T_{dm\_ATC\_DTC}$ (i.e., the MAE of
$T_{dm\_ATC\_four}$ minus the MAE of $T_{dm\_ATC\_DTC}$) and their dependence on $\Delta DTR$ for SURFRAD and FLUXNET data (Fig. 11c &
Fig. 11d). The assessments show that $\Delta DTR$ is generally less than 10 K; and the accuracy of $T_{dm\_ATC\_DTC}$ is better than that of
$T_{dm\_ATC\_four}$. However, with the increase of $\Delta DTR$, the overall accuracy of $T_{dm\_ATC\_four}$ can be superior to $T_{dm\_ATC\_DTC}$. For





SURFRAD data, the overall accuracy of $T_{dm\_ATC\_four}$ is better than that of $T_{dm\_ATC\_DTC}$ once $\Delta DTR$ exceeds 22.0 K (i.e., the $\Delta DTR$ threshold is 22.0 K), while this threshold is 13.0 K for FLUXNET data. With the further increase of $\Delta DTR$, the accuracy of $T_{dm\_ATC\_DTC}$ can be even lower than that of $T_{dm\_ATC\_four}$, e.g., by up to 2.0 K in Fig. 11c and Fig. 11d. In other words, $T_{dm}$ can be estimated more accurately with $T_{dm\_ATC\_four}$ than $T_{dm\_ATC\_DTC}$ once $\Delta DTR$ is relatively large (i.e., Scenario #3).

Note that the optimal threshold of $\Delta DTR$ for the SURFRAD data (22.0 K) differs from that for the FLUXNET data (13.0 K). Here, we set the $\Delta DTR$ threshold as 20.0 K, which is close to that determined for the SURFRAD data, mostly because of the following factors: (1) the SURFRAD sites have been managed uniformly by NOAA (National Oceanic and Atmospheric Administration) for over 15 years, and the associated radiance measurements have been consistently quality-controlled (Augustine et al., 2000); and (2) the land cover types of the SURFRAD sites are not limited to vegetation. We acknowledge

that using a single threshold of 20.0 K may not be optimal for all climate zones and land cover types across the globe, but using of a single threshold effectively avoids outliers due to failed simulations while keeping the simplicity in the global generation of $T_{dm}$ products.

With the thresholds given as above, we provide the percentage of each scenario within each 10-degree latitude zone (Fig. 12). In low- and mid-latitude regions, the percentage of Scenario #2 (i.e., $DTR_{four} \geq 5.0$ K & $\Delta DTR < 20.0$ K) reaches

over 80%, indicating that the IADTC framework mainly uses the DTC-modelled results to estimate $T_{dm}$ in those regions. With the increase of latitude, the percentage of Scenario #1 (i.e., $DTR_{four} < 5.0$ K) gradually increases, mostly due to a decrease in DTR with the weakened incoming solar radiation over higher-latitude regions. The percentage of Scenario #1 reaches around 60% in the Arctic and Antarctic, which echoes well with the small $\Delta T_{sb}$ in high-latitude regions (Fig. 9). The percentage of Scenario #3 (i.e., $DTR_{four} \geq 5.0$ K & $\Delta DTR \geq 20.0$ K) remains relatively stable at around 10% over most

regions across the globe, but it can increase to 20% in the equatorial zone (10 °N ~ 10 °S) and Antarctic, which indicates the relatively poor performance of the DTC model over these regions. The lower performance of the DTC model in the equatorial zone may be related to the high cloud percentage, while over the Antarctic, it reflects the expected difficulties over polar regions (see Section 5.2 for more discussions).





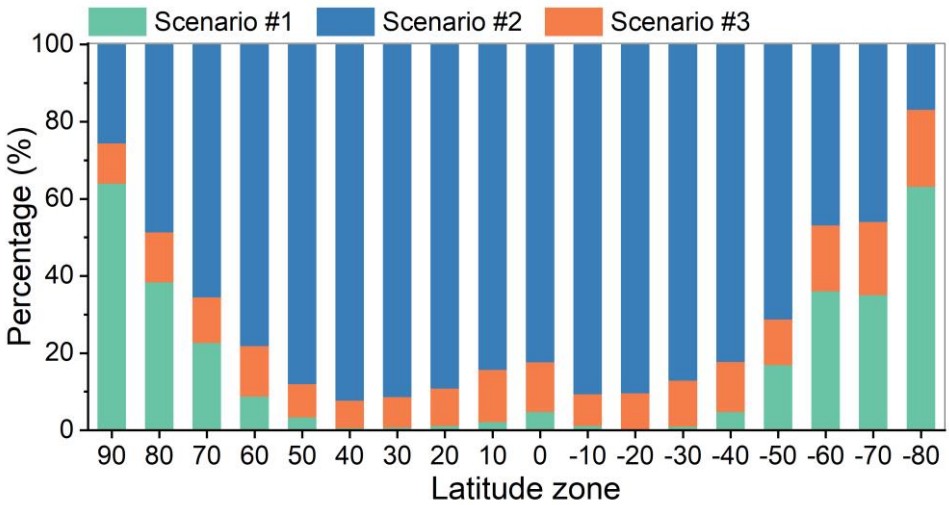


**Fig. 12. Percentage of each scenario (see Fig. 2) within 10-degree latitude intervals. For example, the number '-50' denotes the averaged percentage of each scenario within 50° S to 60° S.**

## 5.2 Possible uncertainty sources of GADTC product

GADTC products uncertainties arise from four main sources: (1) MODIS data quality or LST retrieval error; (2) cloud cover

and contamination; (3) overpass time drift and linear interpolation; and (4) uncertainties associated with the IADTC framework. These four uncertainty sources can affect the under-cloud LST reconstruction with the ATC model as well as the diurnal LST dynamics modelling with the DTC model, and consequently, affect the accuracy of the GADTC products. In addition, these uncertainties can influence each other via error propagation. In the following, we discuss the four error sources and their effect in more detail.

The ATC and DTC models use cloud-free LST observations to estimate $T_{dm}$. Therefore, retrieval errors of MODIS LSTs affect the results of ATC and DTC models and the accuracies of the estimated $T_{dm}$. Fig. A2a shows that the quality of MODIS LSTs in the equatorial regions is lower than that in the other regions. This suggests that GADTC products should have larger uncertainties in equatorial regions where consequently, the IADTC framework may need further improvements.

Cloud percentage can also impact the accuracies of the GADTC products. In regions with a higher cloud percentage,

e.g., the equatorial regions (Fig. A2b), more under-cloud LSTs need to be reconstructed with the ATC model. However, errors of reconstructed under-cloud LSTs are larger than those of cloud-free LSTs. Therefore, regions with a higher cloud percentage are also associated with larger errors from ATC modelling and consequently, DTC modelling and the estimated $T_{dm}$. In polar regions, the cloud detection algorithm has larger uncertainties due to the spectral similarities between clouds and snow (Østby et al., 2014; Westermann et al., 2012), which introduces additional uncertainties to the GADTC products.

The impact of the overpassing time drift mainly occurs over high-latitude regions where the time drift is larger. On the one hand, the cloud-free observations used for solving the free parameters of the ATC model come from significantly



different times within a daily cycle (Fig. A1), which affects the under-cloud LST reconstruction. On the other hand, approximately 30% of the $T_{dm}$ over high-latitude regions were estimated with the DTC modelling results (i.e., Scenario #2; refer to Fig. 12) and the time drift can affect the shape of the DTC curve and, therefore, the estimated $T_{dm}$.

The uncertainties of the GADTC products derived with the IADTC framework mainly include three parts: the reconstruction error of the ATC model, the fitting error of the DTC model, and the choice of the two thresholds. First, the currently used ATC model reconstructs under-cloud LSTs during the day (night) with small positive (negative) biases (Fig. 5), even though information on under-cloud air temperature has been incorporated (Liu et al., 2019b). Second, the DTC model assumes clear-sky conditions and is less capable of simulating under-cloud LST dynamics accurately, which

introduces additional uncertainties especially under some complex situations (Hong et al., 2021). Third, the two fixed thresholds for $DTR_{four}$ and $\Delta DTR$ were determined empirically (Fig. 11): the threshold for $DTR_{four}$ may introduce additional uncertainty over high-latitude regions with small DTRs, while threshold for $\Delta DTR$ may still miss some cases with unrealistic modelling results.

It is difficult to distinguish and quantify the individual contributions of these four uncertainty sources to the estimated

$T_{dm}$, as they can affect the ATC and DTC modelling individually and interactively. We are therefore unable to provide a quality control flag for each pixel of the GADTC products. The validations have shown that the accuracies of the GADTC products are generally acceptable over most areas across the globe. However, there are relatively larger uncertainties over equatorial and polar regions, where further validations of the GADTC products and an optimization of the IADTC framework is required.

**5.3 Future perspectives**

Further improvements of the GADTC product can focus on the following three aspects:

(1) *More extensive validation and inter-comparison of the GADTC products*: The GADTC products have been evaluated with FLUXNET and SURFRAD datasets, which include *in situ* measurements from most climate zones. However, the number of sites is very limited in regions where the uncertainties of the GADTC products are largest (e.g., equatorial and

polar regions; refer to Fig. 1). It is therefore hard to validate the IADTC framework as well as its improvements over these regions, e.g., the use of a multi-type ATC model instead of a single-type ATC model. The current *in situ* data are also insufficient to verify the accuracies of the GADTC products over these regions. It is therefore necessary to obtain more *in situ* measurements over these regions to validate the accuracy of IADTC framework as well as the GADTC product more completely. Furthermore, reanalysis data, which provide long-term spatiotemporally seamless LSTs and have been widely

used in relevant studies (Simmons et al., 2017), can be used to assess the GADTC products (Trigo et al., 2015).

(2) *Rapid generation of high-resolution spatiotemporally seamless $T_{dm}$ product*: Considering the limited computing resource as well as the aim of this study to obtain the spatial distribution of $\Delta T_{sb}$ and LST trends on a global scale, the spatiotemporally seamless daily $T_{dm}$ were generated at a spatial resolution of 0.5 degree. However, current IADTC framework is equally suitable to generate spatiotemporally seamless daily 1-km $T_{dm}$. For local-scale studies, the IADTC





framework can probably be applied directly. While for large-scale (continent-scale or even global-scale) studies or applications, the generation of 1-km spatiotemporally seamless daily $T_{dm}$ could be computationally unaffordable. Under this circumstance, apart from using as many computation resources as possible, we can resort to two strategies to substantially reduce computational complexity.

First, the similarity of the ATC and DTC model parameters among neighboring pixels can be utilized to accelerate the
calculation speed considerably (Hong et al., 2021; Hu et al., 2020; Zhan et al., 2016). Second, the physically-based IADTC framework can also be integrated with some statistical or empirical estimation strategies (both on $T_{dm}$ or on $\Delta T_{sb}$) to help improving the computational efficiency (Xing et al., 2021). This is reasonable as $\Delta T_{sb}$ (and $T_{dm}$) is generally related to local surface properties (Fig. 9 and Fig. 11). For example, for large-scale or global high-resolution generation of spatiotemporally seamless daily 1-km $T_{dm}$, the IADTC framework can be run in some chosen sample regions to obtain adequate training
samples of $T_{dm}$ (or $\Delta T_{sb}$). Based on these samples, statistical relationships between $T_{dm}$ ($\Delta T_{sb}$) and the related variables such as the four daily LSTs, latitude, land cover type, elevation, and cloud percentage can be obtained to help estimate the $T_{dm}$ ($\Delta T_{sb}$) across the globe efficiently. Furthermore, the training samples of $T_{dm}$ ($\Delta T_{sb}$) can also be from geostationary satellite data, which can help reduce the computational complexity of the DTC modelling.

(3) *Generation of $T_{dm}$ with a longer time-span*: The GADTC products can only date back to 2003 because the IADTC
framework requires four observations per day to estimate $T_{dm}$ while MODIS started to provide four daily observations in 2003. However, daily mean LSTs with a longer time-span are strongly required for relevant studies such as climate change analysis (Jin and Dickinson, 2010; Simmons et al., 2017). AVHRR data provide global LST observations before 2000 and recent studies have achieved tremendous progress in the correction of orbit drift in order to generate more accurate AVHRR LST datasets (Julien and Sobrino, 2012; Latifovic et al., 2012; Ma et al., 2020; Liu et al., 2019a). However, the current
IADTC framework is not applicable to AVHRR since it only samples the surface twice per day. It is therefore imperative to develop a framework for $T_{dm}$ estimation that also suits AVHRR-like LSTs. Apart from polar orbiters, geostationary satellites and reanalysis data deliver LST over similar time-spans. Although reanalysis data are still limited by their coarse spatial resolution and geostationary satellite data have a limited spatial coverage, especially over polar regions, the fusion of these datasets has great potential to help generating $T_{dm}$ with a longer time-span (Long et al., 2020; Quan et al., 2018).

**6. Conclusions**

MODIS LST products have been widely used for long-term time series analyses. However, due to the missing LSTs caused by clouds and under-sampling of the diurnal LST dynamics, currently there is still no global dataset of spatiotemporally seamless daily mean LST ($T_{dm}$) with an acceptable systematic sampling bias ($\Delta T_{sb}$), which is caused by averaging only instantaneous cloud-free observations. To resolve this issue, we proposed the IADTC framework by using a more advanced
ATC model as well as by optimizing the estimation of $T_{dm}$ with the DTC model, and generated global spatiotemporally seamless $T_{dm}$ products (i.e., the GADTC products) from 2003 to 2019. Based on SURFRAD and FLUXNET *in situ*





measurements, the IADTC framework was validated with *in situ* measurements at the site level and the GADTC products were directly compared with *in situ* $T_{dm}$ observations. The validations with the SURFRAD and FLUXNET measurements reveal that the IADTC framework is able to reduce the systematic positive sampling bias ($\Delta T_{sb}$) of $T_{dm\_cloud\_free}$, avoid the

outliers caused by failed simulation, and provide relatively accurate estimates of spatiotemporally seamless $T_{dm}$. Based on the GADTC products, we analyzed the global distribution of $\Delta T_{sb}$ and examined the similarities and differences between the GADTC products and $T_{dm\_cloud\_free}$ (daily mean LST based on cloud-free observations).

Our major conclusions are: (1) the validations of the IADTC framework based exclusively on *in situ* measurements at the site level show MAEs of 1.4 K and 1.1 K for the SURFRAD and FLUXNET measurements, respectively; the biases for

these two datasets are both close to zero. (2) The comparisons between the GADTC satellite products and *in situ* $T_{dm}$ observations show that the MAEs for the SURFRAD and FLUXNET measurements are 2.2 K and 3.1 K, respectively; the associated biases for these two datasets are −1.6 K and −1.5 K, respectively. (3) The global mean sampling bias $\Delta T_{sb}$ is 1.8 K, it is usually larger than 2.0 K over low- and mid-latitude regions and close to zero over high-latitude regions. (4) Global mean LST trends derived with the GADTC product and the traditional direct-averaging method are similar (both between

0.025 to 0.029 K/year from 2003 to 2019), while the pixel-based MAE in LST trend derived with these two methods is 0.012 K/year. Despite its limitations, the proposed IADTC framework allows the practical generation of global spatiotemporally seamless $T_{dm}$ and provides insights for generating global long-term high-resolution (e.g., 1km) $T_{dm}$ products. The generated GADTC product should be helpful for relevant applications such as climate change analysis and thermal environment investigations.

**Appendix A. Statistics on the original MODIS MXDC1 V6 products**

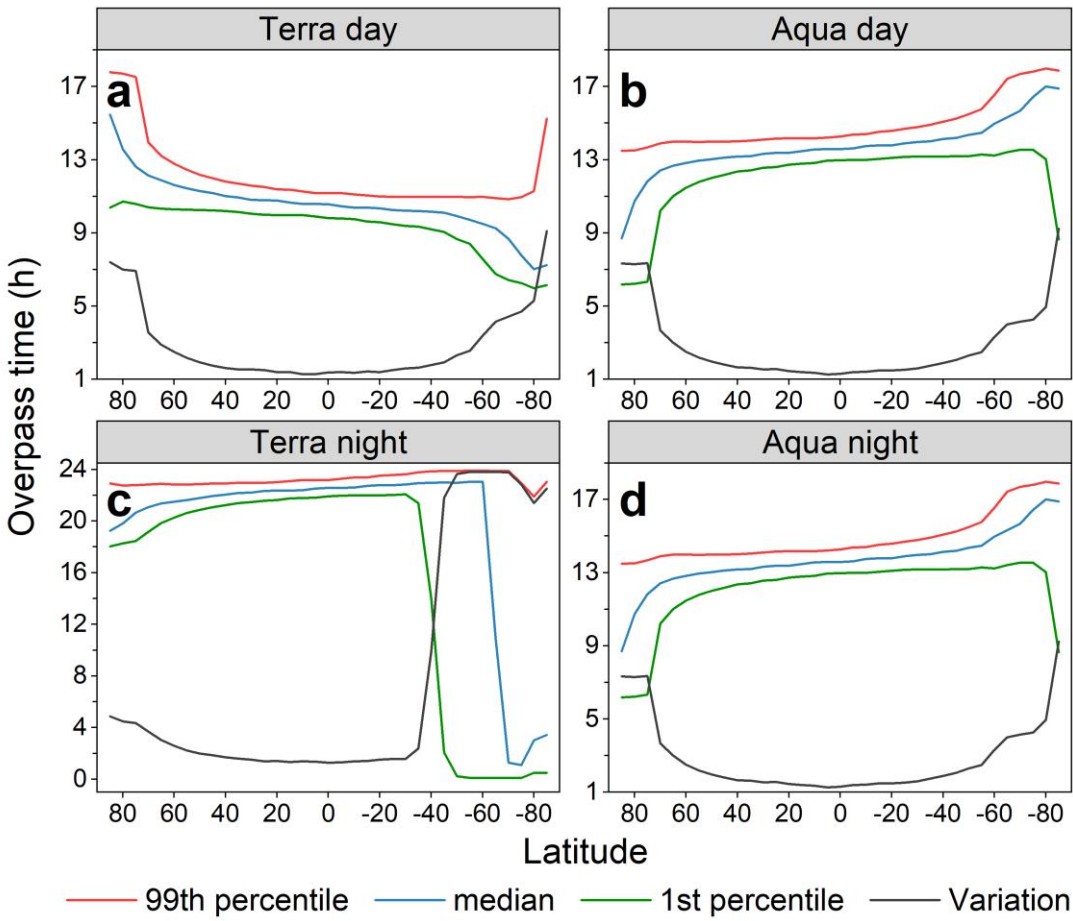

**Fig. A1. Statistics on each MODIS overpassing time within a 10-degree interval from 2003 to 2019. Each subplot displays the 99th percentile, median, 1st percentile and the associated variation (the 99th percentile minus 1st percentile).**

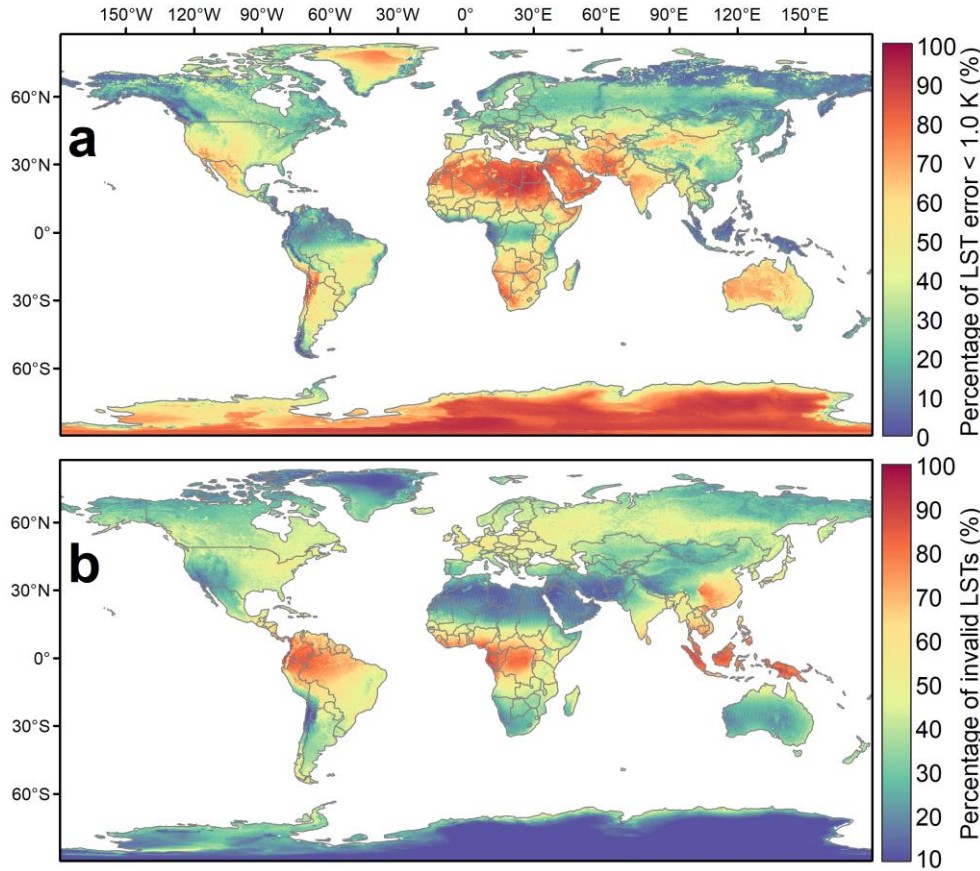

**Fig. A2. Uncertainties of the downloaded MODIS MXD11C1 V6 LSTs. (a) shows the percentage of LSTs with a retrieval error less than 1.0 K; and (b) displays the percentage of invalid data (≈ clouds).**

**Appendix B. Mean absolute errors of $T_{dm\_IADTC}$ and $T_{dm\_OADTC}$ in Scenarios #1 and #3 at the site level**



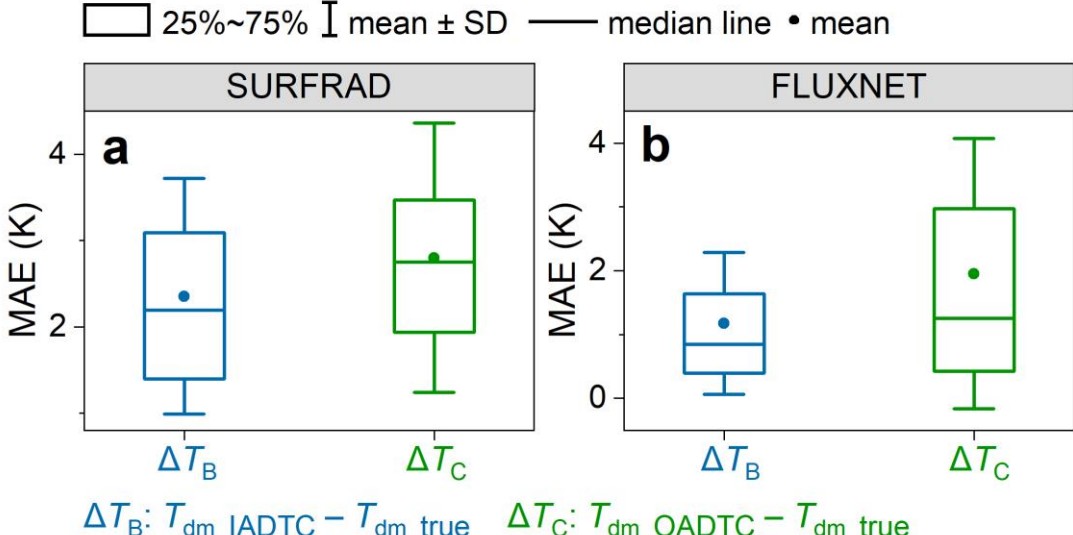

**Fig. B1. Boxplots for the MAEs of the IADTC framework ($T_{dm\_ATC\_DTC}$) and the OADTC framework ($T_{dm\_OADTC}$) under Scenarios #1 and #3. (a) and (b) are for the SURFRAD and FLUXNET measurements, respectively.**




## Appendix C. Distribution of average sampling bias per season







**Fig. C1. Average sampling bias $\Delta T_{sb}$ for indicated three-month interval between 2003 and 2019. (a) displays the $\Delta T_{sb}$ for December-January-February (DJF) and (b) displays the corresponding results averaged over 5-degree intervals**

**longitude. Similarly, (c) and (d), (e) and (f), and (g) and (h) display the corresponding results for March-April-May (MAM), June-July-August (JJA), and September-October-November (SON), respectively.**

**Appendix D. Nomenclature**

**Acronyms**

|  |  |
|---|---|
| ASTER | Advanced Spaceborne Thermal Emission and Reflection Radiometer |
| 600 ATC | annual temperature cycle |
| AVHRR | Advanced Very High-Resolution Radiometer |
| BSV | Barren Sparse Vegetation |
| CRO | Croplands |
| DBF | Deciduous Broadleaf Forests |
| 605 DOY | day of year |
| DTC | diurnal temperature cycle |
| DTR | daily temperature range |
| EBF | Evergreen Broadleaf Forests |
| ENF | Evergreen Needleleaf Forests |
| 610 GADTC product | Global daily mean LST product generated with the improved ADTC-based framework |
| GOES | Geostationary Operational Environmental Satellite |
| GRA | Grasslands |
| IADTC framework | improved ADTC-based framework |
| IGBP | International Geosphere–Biosphere Programme |
| 615 LST | land surface temperature |
| MAE | mean absolute error |
| MERRA-2 | Modern-Era Retrospective analysis for Research and Applications version 2 |
| MF | Mixed Forests |
| MODIS | Moderate Resolution Imaging Spectroradiometer |
| 620 MSG-SEVIRI | the Spinning Enhanced Visible and Infrared Imager onboard Meteosat Second Generation |
| OADTC framework | original ADTC-based framework |
| OSH | Open Shrublands |
| SAV | Savannas |
| SAT | surface air temperature |





SNO                   Snow and Ice

SURFRAD               Surface Radiation Budget Network

WET                   Permanent Wetlands

WSA                   Woody Savannas

**Symbol representation**

$DTR_{four}$           diurnal temperature range calculated by the four LSTs which include the cloud-free LSTs and ATC-reconstructed LSTs

$DTR_{DTC}$            diurnal temperature range calculated by the DTC model

$\Delta DTR$           the difference between $DTR_{DTC}$ and $DTR_{four}$

$T_{dm}$               daily mean LST

$T_{dm\_ATC\_DTC}$     daily mean LST calculated by frequently sampling diurnal LST dynamics modelled by DTC model with cloud-free LST observations and under-cloud LSTs reconstructed by ATC model

$T_{dm\_ATC\_four}$    daily mean LST calculated by averaging cloud-free LST observations and under-cloud LSTs reconstructed by ATC model

$T_{dm\_cloud\_free}$  daily mean LST calculated by averaging cloud-free LST observations

$T_{dm\_IADTC}$        daily mean LST estimated with the IADTC framework

$T_{dm\_OADTC}$        daily mean LST estimated with the OADTC framework

$T_{dm\_true}$         true daily mean LST for validation

$T_{in\_ATC}$          instantaneous under-cloud LSTs reconstructed by ATC model

$T_{in\_ATC\_DTC}$     diurnal LST dynamics modelled by DTC model with cloud-free LST observations and under-cloud LSTs

645                         reconstructed by ATC model

$T_{in\_cloud\_free}$  instantaneous cloud-free LST observations

$T_{in\_obs}$          hourly LST observations

$T_{in\_under\_cloud}$ instantaneous under-cloud LST observations

$\Delta T_{sb}$        sampling bias


**Data availability**

The generated GADTC products are organized yearly and freely available at https://doi.org/10.5281/zenodo.6287052 (Hong et al., 2022). Each file contains the global day-to-day spatiotemporal seamless daily mean land surface temperature, which can be acquired by scaling 0.01 in Kelvin unit.



**Earth System Science Data Discussions**

**Author contributions**

**Falu Hong**: Conceptualization, Methodology, Software, Formal analysis, Writing – original draft preparation, and Writing – Review & Editing. **Wenfeng Zhan**\*: Conceptualization, Methodology, Writing – Review & Editing, Supervision, Project administration, and Funding acquisition. **Frank-M. Göttsche**: Writing – Review & Editing. **Zihan Liu**: Writing – Review & Editing. **Pan Dong**: Writing – Review & Editing. **Huyan Fu**: Writing – Review & Editing. **Fan Huang**: Resources, Writing 660 – Review & Editing. **Xiaodong Zhang**: Writing – Review & Editing.

**Competing interests**

The authors declare that they have no known competing financial interests or personal relationships that could have appeared to influence the work reported in this paper.

**Acknowledgements**

This work is jointly supported by the National Natural Science Foundation of China under Grant 42171306 and the Jiangsu Provincial Natural Science Foundation under Grant BK20180009. We are also grateful for the financial support provided by the National Youth Talent Support Program of China.

The authors also wish to thank the following organizations for providing the data to support this study, including (1) Global Radiation group of Earth System Research Laboratory Global Monitoring Division managed by the National Oceanic 670 and Atmospheric Administration (NOAA) for providing SURFRAD data; (2) Land Processes Distributed Active Archive Center (LP DAAC) managed by the National Aeronautics and Space Administration (NASA) Earth Science Data and Information System (ESDIS) project for providing MOD11C1 and MYD11C1 products; (3) FLUXNET Network hosted by the Lawrence Berkeley National Laboratory for providing the FLUXNET2015 Dataset; and (4) NASA's Goddard Space Flight Center for providing MERRA-2 data.

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
