# Peer review of "A global dataset of spatiotemporally seamless daily mean land surface temperatures: generation, validation, and analysis"

_Earth System Science Data, 2022_

## Author Comment (AC3)

| 1  | Responses to the Manuscript essd-2022-83 RC3:                                 |
|----|--------------------------------------------------------------------------------------|
| 2  | A global dataset of spatiotemporally seamless daily mean land                        |
| 3  | surface temperatures: generation, validation, and analysis                           |
| 4  |                                                                                      |
| 5  | Dear reviewer #3,                                                                    |
| 6  |                                                                                      |
| 7  | The authors would like to thank you for providing us with thoughtful and outstanding |
| 8  | comments. We have addressed all comments in detail and revised the manuscript        |
| 9  | accordingly and tracked the changes so that you can see that we have rewritten many  |
| 10 | parts of the manuscript. Point-by-point responses are provided below.                |
| 11 |                                                                                      |
| 12 | Yours sincerely,                                                                     |
| 13 | Falu Hong, Wenfeng Zhan*, Frank-M. Göttsche, Zihan Liu, Pan Dong, Huyan Fu, Fan      |
| 14 | Huang, and Xiaodong Zhang                                                            |
| 15 |                                                                                      |
| 16 | Email: zhanwenfeng@nju.edu.cn                                                 |
| 17 |                                                                                      |
| 18 |                                                                                      |

| 19 | I. TABLE OF CONTENTS          |
|----|-------------------------------|
| 20 | I. TABLE OF CONTENTS          |
| 21 | II. ATTENTIONS                |
| 22 | III. RESPONSES TO REVIEWER #3 |
| 23 | Comment #14                   |
| 24 | Comment #24                   |
| 25 | Comment #34                   |
| 26 | Comment #45                   |
| 27 | Comment #5                    |
| 28 | Comment #69                   |
| 29 | Comment #79                   |
| 30 | Comment #811                  |
| 31 | Comment #914                  |
| 32 | Comment #1014                 |
| 33 | Comment #1115                 |
| 34 | IV. REFERENCES                |
| 35 |                               |
| 36 |                               |

**38 **II. ATTENTIONS**

- 39 (1) In the following responses, texts contained within the red braces {...} are identical
  40 to those in our revised manuscript.
- 41 (2) In the following responses, the line numbers [Line XXX-XXX] refer to the clean
- 42 version of the revised manuscript.
- 43 (3) Fig. 1, 2, and  $\underline{3}$ ..., and  $\underline{Eq. 1}$ , 2, and  $\underline{3}$ ... refer to the figures and equations
- 44 excerpted from our revised manuscript.
- 45 (4) In the following responses, all the related references are provided collectively in46 Part IV References.
- 47

**48 **III. RESPONSES TO REVIEWER #3**

**49 **Comment #1**

- 50 spatiotemporally seamless land surface temperature at daily, monthly, and yearly
- 51 scales are important for LST-related researches. This study presents a meaningful
- 52 study with the use of MODIS LST product and reanalysis data to generate the mean
- 53 LST value at different scales. It was well organized and the results were with good
- 54 accuracy. Overall, the manuscript can be accepted with minor revision:

55 Authors' reply:

- 56 Thanks for your appreciation. The point-to-point responses are given as follows.
- 57

**58 Comment #2**

- 59 There are many other reanalysis data available and why you choose the MERRA2
- 60 *dataset? What is advantage of this dataset?*

**61 Authors' reply:**

- 62 Thanks for your comment. We agree with you that there are many other
- 63 reanalysis data, such as ERA-land (Muñoz-Sabater et al., 2021), GLDAS (Rodell et
- 64 al., 2004), JRA-55 (Kobayashi et al., 2015), and NCEP (Kalnay et al., 1996)
- 65 reanalysis datasets. We chose MERRA2 dataset because it can provide global hourly
- air temperature. The MERRA2 air temperature can provide the annual air temperature
- 67 variation pattern to simulate LST fluctuations induced by synoptic conditions. This
- 68 information is used in the ATC model to reconstruct the under-cloud LSTs at four
- 69 overpassing times. Other reanalysis datasets can replace the MERRA2 dataset if they
- 70 could provide similar information.
- 71

**72 **Comment #3**

73 The key steps are suggested to be clarified in in figure 2. The pre-processing is not

- 74 *included in this flowchart.*
- 75 Authors' reply:

76 Thanks for your comment. We have added the preprocessing steps which include 77 unifying the projection system and resampling the datasets to the same spatial 78 resolution in the flowchart. The revised flowchart is given as follows for your 79 convenience.

- 81
- 82 Fig. 1. Flowchart of the IADTC framework. *DTR*four refers to diurnal
- 83 temperature range (DTR) calculated as the maximum minus the minimum from
- 84 the gap-free LSTs at the four overpassing times; *DTR*DTC refers to the DTR
- 85 calculated from the hourly LSTs modelled with the DTC model.  $\Delta DTR$  refers to
- 86 the absolute difference between *DTR*four and *DTR*DTC.
- 87

**88 **Comment #4**

- 89 175: A basic equation of the single-type and multi-type model is better to be provided
- 90 *here*.

**91 Comment #5**

92 Figure 3: multi-type ATC models are identical? Why there is no differences? It will be

80

93 a little confused on the naming of the ATC models for single or multi-type model and

94 single or double-sinusoidal ATC model?

95 Authors' reply:

96 Thanks for your comment. Comments #4 and #5 are both related to descriptions
97 of ATC model, so we combine the response. We agree with you that some of the ATC
98 model descriptions are redundant and could be misleading.

We summarized the basic equation of ATC model as Eq. R1. For the single-type ATC model, *M* equals 1 for the global application, i.e., the single-sinusoidal version was applied to the global scale. As for the multi-type ATC model, the value of *M* is different at different latitude zones. In low-latitude (23.5° N – 23.5° S) and highlatitude regions (66.5° N/S – 90° N/S), *M* equals 2, i.e., the double-sinusoidal version was applied to these regions. In mid-latitude regions (23.5° N/S – 66.5° N/S), *M* equals 1, i.e., single-sinusoidal version was used.

106 To address your question about the identical results between the single-type and 107 multi-type ATC models, the results of single-type and multi-type ATC models are 108 identical in mid-latitude region because they both use the single-sinusoidal version (M109 = 1). Therefore, the results in Fig. 3b are identical. While the results of single-type 110 and multi-type ATC models are different in low-latitude and high-latitude regions (Fig. 3a & Fig. 3c) because the single-type ATC model still uses the single-sinusoidal 111 112 version (M=1) while the multi-type ATC model use the double-sinusoidal version (M113 = 2).

114
$$\begin{cases} T_{\text{ATCM}}(d) = T_0 + \sum_{m=1}^{M} A_m \sin\left(\frac{2\pi m d}{N} + \theta_m\right) + k \cdot \Delta T_{\text{air}}(d) \\ \Delta T_{\text{air}}(d) = T_{\text{air}}(d) - T_{\text{ATCO}}(d) \\ T_{\text{ATCO}}(d) = T_0' + \sum_{m=1}^{n} A_m' \sin\left(\frac{2\pi m d}{N} + \theta_m'\right) \end{cases}$$
Eq. R1

115 where  $T_{\text{ATCM}}(d)$  denotes the daily LST variations simulated with the ATC model; M is 116 the number of used harmonic components; d and N are the day of year (DOY) and 117 number of days in a year, respectively;  $\Delta T_{air}(d)$  is the difference between the daily 118 SATs (i.e.,  $T_{air}(d)$ , obtained from MERRA2 reanalysis data) and the modelled air 119 temperatures with the original ATC model ( $T_{\text{ATCO}}(d)$ ); and  $T_0, A_m, \theta_m$ , and k are the 120 parameters that need to be solved with the cloud-free daily LSTs and SATs, usually 121 through the least-square method. 122 To reduce the redundancy and clarify the description, we have revised Section

123 3.1.2. The revised version is given as follows for your convenience.

**125 **3.1.2 Under-cloud LST reconstruction with multi-type ATC model**

- 126 The general formula of ATC model is displayed in Eq. 2. The single-type ATC model 127 in the OADTC framework uses a single sinusoidal function (M = 1 in Eq. 2) to model
- 128 the intra-annual LST variations driven by solar radiation change and incorporates
- 129 surface air temperatures to help simulate the LST fluctuations induced by synoptic
- 130 conditions (Zou et al., 2018; Liu et al., 2019b). The use of a single sinusoidal function
- 131 is generally acceptable for mid-latitude regions. However, a single sinusoidal is no
- 132 longer suitable for low-latitude because there are two solar radiation peaks within a
- 133 yearly cycle over low-latitude regions (Xing et al., 2020; Bechtel, 2015; Cao and
- 134 Sanchez-Azofeifa, 2017); it is also inadequate for high-latitude regions where polar
- 135 days and nights occur (Østby et al., 2014; Liu et al., 2019; Westermann et al., 2012).
- 136 Therefore, the use of the single-type ATC model in the OADTC framework is less
- 137 suitable to generate  $T_{dm}$  at the global scale (Fig. 2). To overcome this limitation, the
- 138 IADTC framework uses different versions of ATC model (termed the multi-type ATC
- 139 model) to reconstruct under-cloud LSTs over the low-, mid-, and high-latitude
- 140 regions, respectively. The details are given as follows:
- 141 (1) Low-latitude regions  $(23.5^{\circ} N 23.5^{\circ} S)$
- 142 The solar radiation possesses two peaks within a yearly cycle over low-latitude
- 143 regions (Fig. 2a). We therefore employed the ATC model with two sinusoidal
- 144 functions (M = 2 in Eq. 2) to reconstruct the daily LST dynamics within an annual
- 145 cycle (Liu et al., 2019b; Xing et al., 2020).
- 146 *(2) Mid-latitude regions (23.5° N/S 66.5° N/S)*
- 147 The solar radiation peaks once in summer during an annual cycle. We therefore
- 148 employed the ATC model with single-sinusoidal function (M = 1 in Eq. 2) to
- 149 reconstruct the daily LST dynamics (Fig. 2b).
- 150 (3) High-latitude regions  $(66.5^{\circ} N/S 90^{\circ} N/S)$
- 151 The polar day/night phenomena occur over high-latitude regions and the duration
- 152 increases with the latitude. Theoretically, over these regions, the ATC model with
- 153 multiple sinusoidal functions should be the best choice. However, the number of
- 154 cloud-free MODIS observations is limited, and additional model complexity can lead
- 155 to over-fitting and weaken the generalization ability of the ATC model (Liu et al.,
- 156 2019b). To balance model accuracy and generalization ability, the ATC model with
- 157 two sinusoidal functions was selected for high-latitude regions (see Fig. 2c).

158
$$\begin{cases} T_{\text{ATCM}}(d) = T_0 + \sum_{m=1}^{M} A_m \sin\left(\frac{2\pi m d}{N} + \theta_m\right) + k \cdot \Delta T_{\text{air}}(d) \\ \Delta T_{\text{air}}(d) = T_{\text{air}}(d) - T_{\text{ATCO}}(d) \\ T_{\text{ATCO}}(d) = T_0' + \sum_{m=1}^{M} A'_m \sin\left(\frac{2\pi m d}{N} + \theta'_m\right) \end{cases}$$
(2)

159 where  $T_{\text{ATCM}}(d)$  denotes the daily LST variations simulated with the ATC model; *M* is 160 the number of used harmonic components; *d* and *N* are the day of year (DOY) and 161 number of days in a year, respectively;  $\Delta T_{\text{air}}(d)$  is the difference between the daily 162 SATs (i.e.,  $T_{\text{air}}(d)$ , obtained from MERRA2 reanalysis data) and the modelled air 163 temperatures with the original ATC model ( $T_{\text{ATCO}}(d)$ ); and  $T_0$ ,  $A_m$ ,  $\theta_m$ , and *k* are the 164 parameters that need to be solved with the cloud-free daily LSTs and SATs, usually 165 through the least-square method.

---

## Author Response (AR1)

**Responses to the Manuscript essd-2022-83:**

**A global dataset of spatiotemporally seamless daily mean land surface temperatures: generation, validation, and analysis**

Dear tropical editor and reviewers,

We submit the revised version of our manuscript (**essd-2022-83**).

The authors would like to thank you and the reviewers for providing us with thoughtful and outstanding comments. We have addressed all comments in detail and revised the manuscript accordingly and tracked the changes so that you can see that we have rewritten many parts of the manuscript. Point-by-point responses to all reviewer remarks are provided below.

We will be very glad to receive your feedback.

Yours sincerely,

Falu Hong, Wenfeng Zhan*, Frank-M. Göttsche, Zihan Liu, Pan Dong, Huyan Fu, Fan Huang, and Xiaodong Zhang

Email: zhanwenfeng@nju.edu.cn

**I. TABLE OF CONTENTS**

**II. ATTENTIONS**

(1) In the following responses, texts contained within the red braces {…} are identical
to those in our revised manuscript.

(2) In the following responses, the line numbers [Line XXX-XXX] refer to the clean
version of the revised manuscript.

(3) Fig. 1, 2, and 3…, and Eq. 1, 2, and 3… refer to the figures and equations
excerpted from our revised manuscript.

(4) In the following responses, all the related references are provided collectively in
Part VI References.

 **III. RESPONSES TO REVIEWER #1**

**Comment #1**

*This study designed an operational framework that uses the annual temperature cycle*

*(ATC) and diurnal temperature cycle (DTC) models to generate global seamless daily*

*mean land surface temperature (LST). The framework and generated product were*

*validated with globally distributed in situ measurements. The validations show that*

*the generated daily mean LST can correct the sampling bias caused by directly*

*compositing the cloud-free MODIS LSTs. This is an interesting point for the thermal*

*remote sensing community. Additionally, the authors discussed the uncertainties of the*

*daily mean LST products, which are useful for further improvement. The authors*

*clearly addressed the structure of the IADTC framework and comprehensively*

*evaluated the generated daily mean LST product. This manuscript is generally well*

*written and clearly organized. I recommend the paper for publication after the*

*following issues are answered.*

**Authors' reply:**

Thanks very much for your appreciation. We have provided the point-to-point response to the concerned issues below.

**Major comments**

**Comment #2**

*The direct comparison results between the generated daily mean land surface*

*temperature product and in situ measurements display systematically negative bias at*

*most sites (Tables 1 and 2). The authors should provide more explanations about the*

*negative bias.*

**Authors' reply:**

Thanks for your comment. The systematically negative bias between the *in situ*

measurement and GADTC product is directly related to the systematic negative bias between instantaneous *in situ* measurement and instantaneous MODIS land surface temperature (LST) observations. The comparison results between instantaneous

SURFRAD LST and MODIS LST observations (Fig. R1) show that the mean bias is negative at four overpassing times. Since the GADTC products are generated based on the instantaneous MODIS LST observations, the systematically negative bias within the instantaneous observations will be propagated to the generated daily mean LST.

The systematically negative bias between the instantaneous MODIS LST observations and *in situ* measurements could be caused by: (1) the spatial mismatch between the satellite and *in situ* measurement; (2) the differences in the observation angles; (3) the uncertainties from the LST retrieval algorithm, such as the estimation of broadband emissivity (Guillevic et al., 2018).

To avoid those uncertainties and fully reflect the accuracy of IADTC framework, we validated the IADTC framework with single source *in situ* measurements (Figs. 6 & 7). Results show that the MAEs of the IADTC framework are 1.4 K and 1.1 K for SURFRAD and FLUXNET data, respectively; and the mean biases are both close to zero.

[Figure]

Fig. R1. Comparison between the SURFRAD instantaneous observations and MODIS instantaneous observations for the Terra day (**a**), Aqua day (**b**), Terra night (**c**), and Aqua night (**d**) overpassing times.

**Comment #3**

*The authors used the diurnal temperature range (DTR) to define different scenarios.*

*In this paper, the calculated DTR can be affected by the accuracy of ATC model, then*

*affecting the determination of which scenario is used to generate daily mean land*

*surface temperature. I recommend the authors add more discussions about the*

*uncertainties of ATC model to the daily mean LST estimation.*

**Authors' reply:**

Thanks for your comment. We agree with you that the accuracy of the ATC

model can affect the determination of scenarios. We compared the proportion of three scenarios using the ATC-reconstructed under-cloud LSTs and actual *in situ* undercloud LST observations based on the SURFRAD and FLUXNET datasets, respectively (Table R1). Table R1 proves that the accuracy of ATC model can affect the determination of scenarios. We have added discussions about the uncertainties of

ATC model to the scenario determination and $T_{dm}$ estimation in Line 504-507, which was give as follows for your convenience.

Line 504-507:

{First, the currently used ATC model reconstructs under-cloud LSTs during the day (night) with small positive (negative) biases (**Error! Reference source not**

**found.**), even though information on under-cloud air temperature has been incorporated (Liu et al., 2019b). Additionally, the errors in the ATC model can affect the determination of scenarios and consequently, the way to calculate the $T_{dm}$.}

Table R1. The percentage of each scenario using ATC-reconstructed under-cloud LST

and actual *in situ* under-cloud observations for the SURFRAD and FLUXNET

datasets.

| | | Scenario #1 | Scenario #2 | Scenario #3 |
|---|---|---|---|---|
| SURFRAD | $T_{ins\_cloud\_free}$ + $T_{ins\_ATC}$ | 0.2% | 95.0% | 4.8% |
| | $T_{ins\_cloud\_free}$ + $T_{ins\_obs}$ | 7.3% | 86.5% | 6.3% |
| FLUXNET | $T_{ins\_cloud\_free}$ + | 10.1% | 82.5% | 7.3% |

| | | | |
|---|---|---|---|
| $T_{\text{ins\_ATC}}$ | | | |
| $T_{\text{ins\_cloud\_free}} +$ $T_{\text{ins\_obs}}$ | 21.1% | 67.1% | 11.8% |

**156**

**157** **Minor comments**

**158** **Comment #4**

**159** *Line 138: I recommend the authors to add some descriptions about how they process*

**160** *the in situ measurement outliers.*

**161** **Authors' reply:**

**162** Thanks for your comment. We have added the descriptions of processing the

**163** outliers within the *in situ* measurement. Firstly, the minutely or half-hourly

**164** observations were aggregated into hourly values to reduce the impact from short-term

**165** LST fluctuations. Secondly, the outliers in the *in situ* measurements were further

**166** filtered using the '3σ-Hampel identifier' when validating the GADTC products

**167** (Zhang et al., 2020; Göttsche et al., 2016). You can refer to Line 139-140 and Line

**168** 299-302 for reference, which are given as follows for your convenience.

**169** Line 139-140:

**170** {To reduce the impacts of short-term LST fluctuations on validation, we

**171** aggregated minutely observations into hourly values.}

**172** Line 299-302:

**173** {Note that outliers in the *in situ* measurements were removed before performing

**174** the accuracy evaluation; here outliers are defined as the $T_{\text{dm}}$ differences between *in*

**175** *situ* measurements and GADTC products deviating by more than 3σ (three standard

**176** deviations) from the mean (Göttsche et al. 2016; Zhang et al., 2020).}

**177**

**178** **Comment #5**

**179** *Line 176-178: Please add more examples or references about the LST change in low-*

**180** *latitude and high-latitude regions.*

**181** **Authors' reply:**

**182** Thanks for your comment. We have added the references which describe the LST

**183** change in low-latitude (Cao and Sanchez-Azofeifa, 2017) and high-latitude regions

**184** (Østby et al., 2014; Westermann et al., 2012). Please refer to Line 177-180, which is given as follows for your convenience.

Line 177-180:

{However, a single sinusoidal is no longer suitable for low-latitude because there are two solar radiation peaks within a yearly cycle of low-latitude regions (Xing et al.,

2020; Bechtel, 2015; Cao and Sanchez-Azofeifa, 2017); it is also inadequate for high- latitude regions where polar days and nights occur (Østby et al., 2014; Liu et al.,

2019; Westermann et al., 2012).}

**Comment #6**

*Line 218: Temporal normalization is a good way to handle the overpassing time*

*fluctuations. Please provide more discussions about the role of temporal*

*normalization in generating consistent LST products.*

**Authors' reply:**

Thanks for your comment. We totally agree with you that temporal normalization is useful for correcting the overpassing time fluctuations and generating consistent

LST products (Ma et al., 2022). We have added the discussions in Line 499-502 to emphasize the role of temporal normalization in reducing the negative impact of overpassing time fluctuation, which was given as follows for your convenience.

Line 499-502:

{Temporal normalization methods can adjust the LST observations at fluctuated overpassing time to the fixed time, which can eliminate the uncertainties in the under- cloud LST reconstruction and diurnal LST dynamics modeling (Ma et al., 2022; Liu et al., 2019; Duan et al., 2014).}

**Comment #7**

*Line 242: Moving this sentence after the introduction of DTRfour would be better.*

**Authors' reply:**

Thanks for your comment. We agree with you that moving the sentence at Line

242 to the position consequent to the introduction of $DTR_{four}$ would be better for understanding. You can refer to Line 235-238 for the revised manuscript, which was given as follows for your convenience.

Line 235-238:

{The first criterion is based on the diurnal temperature range (DTR), which was calculated as the maximum minus the minimum LSTs within a diurnal cycle.

Specifically, the DTR calculated by four LSTs within the diurnal cycle (termed

$DTR_{four}$) was used (Fig. 5). Here these four daily LSTs can consist of both cloud-free observations ($T_{in\_cloud\_free}$, the green circles in Fig. 1) and under-cloud LSTs reconstructed by the ATC model ($T_{in\_ATC}$, the blue triangles in Fig. 1).}

**Comment #8**

*Fig. 4: I recommend the authors to add one subplot for the illustration of Scenario #1.*

**Authors' reply:**

Thanks for your comment. We have added the subplot to illustrate Scenario #1 in

Fig. 4. The corresponding caption was also revised. The revised Fig. 4 and caption are attached as follows for your reference.

[Figure]

**Fig. 1. Estimation of $T_{dm}$ under different conditions. (a) displays an example of**

**estimating $T_{dm}$ by averaging $T_{in\_cloud\_free}$ and $T_{in\_ATC}$ when $DTR_{four}$ is less than 5.0**

**K (i.e., Scenario #1); (b) displays an example of estimating $T_{dm}$ based on the DTC**

**modelling results (i.e., Scenario #2); (c) displays an example of estimating $T_{dm}$ by**

averaging $T_{\text{in\_cloud\_free}}$ and $T_{\text{in\_ATC}}$ when $\Delta DTR$ is equal or greater than 20.0 K (i.e., Scenario #3). The green circles, red rectangles, and blue triangles denote the instantaneous cloud-free LST observations, under-cloud LST observations, and under-cloud LSTs reconstructed by the ATC model, respectively. The black lines denote the *in situ* LST observations while the blue lines show the DTC-modelled values based on the cloud-free LST observations and ATC-modelled under-cloud LSTs. Noting that hours larger than 24 along the *x*-axis correspond to the next day.

**Comment #9**

*Line 317: "Lower accuracy" being compared to what needs to be clarified.*

**Authors' reply:**

Thanks for your comment. "Lower accuracy" was compared to the accuracy of $T_{\text{dm\_IADTC}}$. This sentence indicates that the accuracy of $T_{\text{dm\_cloud\_free}}$ is lower than that of $T_{\text{dm\_IADTC}}$. It has been revised for clarification. Please refer to Line 319-320 for reference, which was given as follows for your convenience.

Line 319-320:

{By contrast, the MAEs of the $T_{\text{dm\_cloud\_free}}$ are 4.1 K and 2.5 K at the daily and monthly scales, respectively, i.e., they indicate a significantly lower accuracy compared to that of $T_{\text{dm\_IADTC}}$.}

**Comment #10**

*Line 394: Please provide more evidence about the link between ΔTsb and land cover type or DTR.*

**Authors' reply:**

Thanks for your comment. We acknowledge that our original description could be misleading and have clarified the statement with more references cited. Please refer to Line 397-400, which is given as follows for your convenience.

Line 397-400:

{We further observe that $\Delta T_{\text{sb}}$ is sensitive to land cover type and that DTR can partially explain $\Delta T_{\text{sb}}$. For instance, regions with a large DTR (e.g., deserts or bare soils) usually have a greater $\Delta T_{\text{sb}}$ (Sharifnezhadazizi et al., 2019; Hong et al., 2021;

Jin and Dickinson, 2010).}

**Comment #11**

*Line 414: Please clarify what's the different information contained within the ΔTsb.*

**Authors' reply:**

  Thanks for your comment. We are sorry for causing the misunderstanding. This sentence wants to claim that the slope difference between $T_{dm\_cloud\_free}$ and $T_{dm\_IADTC}$

was related to the variation of $\Delta T_{sb}$, and the variation of $\Delta T_{sb}$ is related to the cloud percentage and cloud duration among different months. For clarification, we have rephrased the original description. Please refer to Line 418-419, which was given as follows for your convenience.

Line 418-419:

  {The slope difference is related to the variation of $\Delta T_{sb}$, which can be affected by the cloud percentage and cloud duration among different months.}

**Comment #12**

*Fig. 11: I am wondering about the variation of error of Tdm_ATC_DTC versus*

*DTRfour, which can provide more solid support for the necessity of defining Scenario*

*#1.*

**Authors' reply:**

  Thanks for your comment. The variation of the error of $T_{dm\_ATC\_DTC}$ *versus*

$DTC_{four}$ was displayed in Fig. R2. Results show that under scenario #1 (i.e., $DTR_{four} <$

5.0 K), the error of $T_{dm\_ATC\_DTC}$ is close to the error of $T_{dm\_ATC\_four}$, i.e., mostly near zero, which indicates that $T_{dm\_ATC\_DTC}$ and $T_{dm\_ATC\_four}$ can be used interchangeably to achieve similar accuracy. Additionally, defining Scenario #1 can effectively avoid the outliers caused by the failed simulation case of DTC model.

[Figure]

Fig. R2. The variation of $T_{dm\_ATC\_DTC}$ depends on the variation of $DTR_{four}$. (a) and (b) display the results for SURFRAD and FLUXNET, respectively.

**IV. RESPONSES TO REVIEWER #2**

**Comment #1**

*This paper describes an improved annual and diurnal temperature cycle-based framework method to generate global spatiotemporally seamless daily mean LST products from MODIS data with the support of reanalysis data. The developed dataset performs very well against global in-situ surface observations. Overall, this new method produces a 0.5 --degree daily product of daily mean LST over the globe. Given that this data has high spatial resolution at a daily time scale, it should be a useful tool for climate studies after its flaws are addressed.*

**Authors' reply:**

Thanks very much for your appreciation. We have addressed the flaws you mentioned. Please refer to the following point-to-point response for the details.

**Major comments**

**Comment #2**

*The developed GADTC product has a spatial resolution of 0.5-degree, how to deal with the scale mismatch between the in-situ measurements and the product, the validation can be carried out at a higher spatial resolution, such as MODIS original resolution. Maybe, the authors can classify the in-situ sites to different levels according to the spatial heterogeneity of the site, to further analyze the errors at different sites.*

**Authors' reply:**

Thank you for your comment. This comment is related to three issues: (1) addressing the scale mismatch between *in situ* measurements and generated GADTC product; (2) validating the daily mean land surface temperature (LST) product at the MODIS original resolution; (3) analyzing the errors according to the spatial heterogeneity of the sites.

*(1) Addressing the scale mismatch between in-situ measurements and product*

We agree with you that the scale mismatch exists between *in situ* measurement and satellite-based LST product. To avoid the scale mismatch, we validate the framework merely based on *in situ* measurement, i.e., running the IADTC framework with *in situ* measurement and then using hourly measurements for validation. The results in Section 4.1 show that the mean absolute errors (MAEs) of the IADTC framework are 1.4 K and 1.1 K for SURFRAD and FLUXNET data, respectively. The validation results merely based on *in situ* measurements are better than the validation results through comparing *in situ* measurements and the GADTC product which involves the scale mismatch uncertainty.

*(2) Validating the daily mean LST product at the MODIS original resolution*

According to your suggestion, we ran the IADTC framework with the MOD11A1 and MYD11A1 LST products to validate the daily mean LST product at the MODIS original resolution (~1 km). The seven SURFRAD sites in 2019 were used for validation. Table R2 shows the validation results at the MODIS original resolution are comparable with the validation results at 0.5 degree, i.e., MAE around 2.2 K, except for the DRA site where the MAE exceeds 4.5 K at the original resolution. The abnormal larger errors at DRA site have been reported by previous studies which validated the instantaneous LST product (Duan et al., 2019; Ermida et al., 2020). For clarification, the unsuitable descriptions of the validation results at DRA site in the original manuscript (Line 357-359) have been deleted.

Table R2. Validation results at the MODIS original resolution with the seven SURFRAD sites in 2019.

| Site ID | Bias (K) | MAE (K) | RMSE (K) | STD (K) | R-square |
|---------|----------|---------|----------|---------|----------|
| BON | -1.61 | 2.04 | 2.45 | 1.85 | 0.97 |
| TBL | -0.67 | 2.20 | 2.76 | 2.68 | 0.94 |
| DRA | -4.41 | 4.51 | 5.05 | 2.45 | 0.97 |
| FPK | -1.07 | 2.20 | 2.86 | 2.65 | 0.97 |
| GWN | -1.89 | 2.13 | 2.48 | 1.61 | 0.97 |
| PSU | -2.08 | 2.27 | 2.70 | 1.73 | 0.98 |
| SXF | -1.16 | 1.88 | 2.36 | 2.06 | 0.98 |

*(3) Analyzing the errors according to the spatial heterogeneity of the site*

We define the spatial heterogeneity of SURFRAD sites by calculating the standard deviation of the land cover types within the MODIS original resolution pixel footprint (Fig. R3 & Table R3). The land cover types were obtained from the LCMAP collection 1.1 land cover map in 2019 (Brown et al., 2020). Table R3 shows that BON

and TBL sites are relatively homogeneous, and GWN and PSU sites are relatively heterogeneous. However, the validation results are not expected to be related to spatial heterogeneity. This is probably because, at MODIS original resolution (~1 km), the uncertainty of scale mismatch still exists, and other factors, such as the sensor differences and atmosphere correction uncertainties, can also affect the validation results. Due to these concerns, apart from the direct comparison between *in* situ measurement and satellite-based daily mean LST, we also validated the IADTC

framework merely based on *in situ* measurement to avoid the uncertainty of scale mismatch.

[Figure]

Fig. R3. The land cover types of each SURFRAD site within the MODIS 1-km pixel footprint.

Table R3. The standard deviation of the land cover types of each SUFRAD site within the MODIS pixel footprint.

| Site ID | Land cover STD |
|---------|----------------|
| BON | 0.111 |
| TBL | 0.112 |
| DRA | 0.582 |
| FPK | 0.376 |

| | |
|---|---|
| GWN | 0.907 |
| PSU | 0.691 |
| SXF | 0.385 |

**Comment #3**

*The Surfrad site only has 7 sites, Why not merge the data from the Surfrad and Fluxnet networks when validating the Tdm product. Also, in section 5.1, the ΔDTR can be obtained using the Surfrad and Fluxnet data together.*

**Authors' reply:**

Thank you for your comment. We agree with you that the validation results using the merged SURFRAD and FLUXNET datasets should be provided for readers' reference and convenience. Therefore, we added the contents displaying the validation results using the merged SURFRAD and FLUXNET. The updated Fig. 8 and Fig. 11 in the revised manuscript display the validation results of the $T_{dm}$ product and the determination of $\Delta DTR$ with the merged SURFRAD and FLUXNET datasets, which would be given at the end of this reply for your convenience.

However, in the revised manuscript, we still kept the separate validation results because the differences between SURFRAD and FLUXNET networks can also provide valuable information for readers. Their differences were summarized as follows:

(1) Their data sources are different. The SURFRAD sites have been managed uniformly by National Oceanic and Atmospheric Administration (NOAA) for over 15 years, and the associated radiance measurements have been consistently quality-controlled (Augustine et al., 2000). In contrast, FLUXNET sites are managed by different principal investigators. The quality control might not be consistent as SURFRAD sites.

(2) Their observation numbers are unevenly distributed. The number of FLUXNET sites is far more than the number of SURFRAD sites (126 *vs* 7). Consequently, the number of FLUXNET observations is far more than SURFRAD observations (226220 *vs* 42600). If we merged these two datasets, the results would be determined predominantly by FLUXNET dataset which occupies the majority. In other words, the contribution of SURFRAD dataset would be largely ignored.

(3) The covering land cover types are different. FLUXNET sites are mainly

**17 / 42**

located in vegetated areas. In contrast, the land cover types of the SURFRAD sites are not limited to vegetated areas. SURFRAD sites additionally cover barren area (the

DRA site). Merging them would reduce the contributions from diverse land cover types.

[Figure]

**Fig. 2. GADTC products versus *in situ* observations. (a), (b), and (c) compare the**

**daily mean LST over the SURFRAD, FLUXNET, and combined sites,**

**respectively; and (d), (e), and (f) show the corresponding results for monthly**

**mean LST. The biases were calculated by the GADTC products minus the *in situ***

**measurements. The red ellipse in (b) highlights the cases with notably large**

**errors.**

[Figure]

Fig. 3. Threshold determination for the two criteria in Fig. 5. (a), (b), and (c) display the errors of $T_{dm\_ATC\_four}$ ($T_{dm\_ATC\_four}$ minus $T_{dm\_true}$) depending on $DTR_{four}$ for SURFRAD, FLUXNET, and combined data, respectively; and (d), (e), and (f) display the MAE differences between $T_{dm\_ATC\_four}$ and $T_{dm\_ATC\_DTC}$ (i.e., the MAE of $T_{dm\_ATC\_four}$ minus the MAE of $T_{dm\_ATC\_DTC}$) depending on the $\Delta DTR$ for SURFRAD, FLUXNET, and combined data, respectively. The black lines in (d), (e), and (f) denote the averaged MAE difference within every unit along the x-axis.

**Comment #4**

*The authors used MAE and bias, why not use the RMSE, which is typically used in the LST validation.*

**Authors' reply:**

Thank you for your comment. We agree with you that the RMSE results should be included in the LST validation results. The updated Fig. 8, Table 1, and Table 2 are given as follows for your convenience.

[Figure]

**Fig. 4. GADTC products versus *in situ* observations. (a), (b), and (c) compare the daily mean LST over the SURFRAD, FLUXNET, and combined sites, respectively; and (d), (e), and (f) show the corresponding results for monthly mean LST. The biases were calculated by the GADTC products minus the *in situ* measurements. The red ellipse in (b) highlights the cases with notably large errors.**

**Table 1. Validation results obtained over the seven SURFRAD sites.**

| Site ID | Lat./Long. | IGBP | $N^*$ | Bias (K) | MAE (K) | RMSE (K) | STD (K) |
|---------|-----------|------|-------|----------|---------|----------|---------|
| BON | 40.05°/−88.37° | CRO | 6153 | −1.20 | 1.97 | 2.44 | 2.12 |
| TBL | 40.13°/−105.24° | GRA | 6124 | −1.37 | 2.30 | 2.89 | 2.54 |
| DRA | 36.62°/−116.02° | BSV | 6102 | −2.04 | 2.26 | 2.69 | 1.74 |
| FPK | 48.31°/−105.10° | GRA | 6157 | −1.78 | 2.54 | 3.18 | 2.63 |
| GWN | 34.25°/−89.87° | WSA | 6144 | −1.83 | 2.25 | 2.70 | 1.98 |
| PSU | 40.72°/−77.93° | CRO | 6134 | −1.30 | 1.85 | 2.24 | 1.82 |
| SXF | 43.73°/−96.62° | CRO | 5786 | −1.39 | 2.06 | 2.54 | 2.13 |

*: $N$ denotes the number of days used for validation.

**Table 2. Validation results for the GADTC products stratified by IGBP land cover type of the FLUXNET sites.**

| IGBP | Site number | $N^*$ | Bias (K) | MAE (K) | RMSE (K) | STD (K) |
|------|-------------|-------|----------|---------|----------|---------|
| MF | 5 | 7564 | −1.95 | 2.62 | 3.25 | 2.61 |

| | | | | | | |
|---|---|---|---|---|---|---|
| EBF | 11 | 29588 | −1.71 | 2.75 | 3.34 | 2.87 |
| WET | 15 | 14556 | −0.66 | 2.76 | 4.22 | 4.17 |
| DBF | 19 | 32594 | −1.78 | 2.89 | 3.56 | 3.08 |
| SAV | 5 | 10355 | −2.65 | 3.16 | 3.84 | 2.79 |
| CRO | 14 | 14387 | −1.59 | 3.26 | 4.10 | 3.78 |
| GRA | 23 | 45257 | −1.62 | 3.32 | 4.22 | 3.90 |
| ENF | 25 | 58616 | −0.81 | 3.38 | 4.18 | 4.10 |
| WSA | 5 | 7810 | −2.33 | 3.44 | 4.06 | 3.32 |
| OSH | 3 | 5090 | −3.34 | 3.62 | 4.33 | 2.75 |
| SNO | 1 | 403 | −3.39 | 4.80 | 5.91 | 4.84 |

*: *N* denotes the number of days used for validation.

**Minor comments**

**Comment #5**

*Line 67, some latest papers about the C6 MODIS LST accuracy can be added, such as*

*DOI: 10.1109/TGRS.2020.2998945, https://doi.org/10.1016/j.jag.2018.04.006*

**Authors' reply:**

Thank you for your reminder. We have added the reference you mentioned.

**Comment #6**

*Line 104, the MxD11C1 was derived using the day/night algorithm and giving a*

*reference*

**Authors' reply:**

Thanks for your comment. We have added the reference from Wan and Li (1997)

which is the representative study using the day/night algorithm to derive land surface temperature. The revised sentence in Line 103-105 is given as follows for your convenience.

Line 103-105:

{The MODIS LSTs were retrieved with a refined generalized split-window algorithm, and their accuracies are mostly within 1.0 K over homogeneous surfaces (Wan and Li, 1997; Duan et al., 2019; Wan, 2014).}

**Comment #7**

*Line 139, how to get the hourly values?*

**Authors' reply:**

Thank you for your comment. SURFRAD *in situ* measurements can provide minutely observations and FLUXNET *in situ* measurements can provide half-hourly observations (a part of the sites provide hourly observations). To get hourly values, we aggregated minutely or half-hourly observations to hourly values. This step was to reduce the impact of short-term LST fluctuations caused by local weather variation. In

Line 139-140, we mentioned the way how to get the hourly values, which were given as follows for your convenience.

Line 139-140:

{To reduce the impacts of short-term LST fluctuations on validation, we aggregated minutely observations into hourly values.}

**Comment #8**

*Line 319, Scenarios #1 and #3, How many sites per scenario, the results can be*

*analyzed by scenario, not by Surfrad and Fluxnet.*

**Authors' reply:**

Thanks for your comment. We calculated the count and the percentage of each scenario for the SURFRAD and FLUXNET datasets (Table R4). In addition, we provided the accuracy results by scenario (Fig. R4). Table R4 shows that Scenarios

#1, #2, and #3 covers 0.2%, 95.0%, and 4.8% for the SURFRAD datasets, and 10.2%,

82.5%, and 7.3% for FLUXNET datasets. Fig. R4 shows that for SURFRAD dataset, the MAE in Scenario #2 is the smallest, then followed by Scenario #1 and Scenario

#3. For FLUXNET dataset, the order of MAE in each scenario is: Scenario #3 >

Scenario #2 > Scenario #1. For both two datasets, the bias in Scenario #2 is slightly lower than zero, and the biases in Scenarios #1 and #3 are larger than zero. We should note that although the performances of IADTC framework in Scenarios #1 and #3 are not good as the performance in Scenario #2, the IADTC framework stills performs better than the OADTC framework in Scenarios #1 and #3 (refer to Fig. 6 and Fig. B1

in the manuscript).

We have added the descriptions of the percentage of each scenario for

SURFRAD and FLUXNET sites. Please refer to Line 321-323 and Line 345-347, which were given as follows for your convenience. In Fig. B1 in the Appendix section, the MAEs under scenarios #1 and #3 were also provided for reader's convenience.

Line 321-323:

{The proportion of three scenarios were 0.2%, 95.0%, and 4.8%, respectively. In

Scenarios #1 and #3 under which the accuracies were improved compared with the

OADTC framework, the IADTC framework improves the MAE of estimated $T_{dm}$ by around 0.45 K (from 2.80 K to 2.35 K, see Fig. B1a).}

Line 345-347:

{The proportion of each scenario is 10.2%, 82.5%, and 7.3%, respectively.

Compared with the OADTC framework, in Scenarios #1 and #3 (the proportion is

17.4%) under which the accuracies are considerably improved, IADTC framework improved the MAE of the estimated Tdm by around 0.78 K (from 1.95 K to 1.17 K, refer to Fig. B1b).}

Table R4. The count and percentage of each scenario for the SURFRAD and

FLUXNET datasets.

| | | Scenario #1 | Scenario #2 | Scenario #3 |
|---|---|---|---|---|
| SURFRAD | Count | 84 | 40820 | 2076 |
| | Percentage | 0.2% | 95.0% | 4.8% |
| FLUXNET | Count | 19724 | 161095 | 14333 |
| | Percentage | 10.2% | 82.5% | 7.3% |

[Figure]

Fig. R4. Boxplot of errors of $T_{dm\_IADTC}$ for each scenario. (a) and (b) display the boxplot of mean absolute error (MAE) and bias based on SURFRAD dataset, respectively; and (c) and (d) display are the same as (a) and (b), but for FLUXNET dataset.

**Comment #9**

*Line 360, Fig.8, combines data from the two networks.*

**Authors' reply:**

Thank you for your comment. This reply is related to Comment #3. We have added the figures showing the validation results using the combined data from the two networks in the revised Fig. 8.

**Comment #10**

*Line 373, how to prove the large errors at these sites are related to the high spatial heterogeneity*

**Authors' reply:**

Thank you for your comment. We need to clarify that spatial heterogeneity is one of the many possible reasons for causing large errors. Other factors, such as spatial representativeness and erroneous observations can also cause large errors. In Comment #2, the validation results at SURFRAD sites show that the errors could be large in the homogeneous sites, for example, the DRA site.

For the AU-Wac, CH-Fru, SJ-Adv, and US-Orv sites which have the top 4 largest RMSE ($\geq$ 8.0 K) among the selected 126 FLUXNET sites, we have checked their google earth image within the 0.5 × 0.5 degree and found that their observation field is quite different from their located 0.5-degree grids (Fig. R5). Therefore, we speculate that the larger errors at these sites are related to the high spatial heterogeneity. We clarified this point in Line 375-378, which was given as follows for your convenience.

Line 375-378:

{The relatively larger errors at several FLUXNET sites (e.g., AU-Wac, SJ-Adv, and CH-Fru sites, with MAEs larger than 8.0 K; refer to the red ellipse in Fig. 2e) partly account for the lower accuracy. The relatively large errors at these sites might be related to the erroneous *in situ* measurements as well as the high spatial heterogeneity around these sites.}

[Figure]

Fig. R5. The google earth images for the AU-Wac (a), CH-Fru (b), SJ-Adv (c), and

US-Orv (d) sites. The image boundary is around 0.5 by 0.5 degree.

**V. RESPONSES TO REVIEWER #3**

**Comment #1**

*spatiotemporally seamless land surface temperature at daily, monthly, and yearly*
*scales are important for LST-related researches. This study presents a meaningful*
*study with the use of MODIS LST product and reanalysis data to generate the mean*
*LST value at different scales. It was well organized and the results were with good*
*accuracy. Overall, the manuscript can be accepted with minor revision:*

**Authors' reply:**

Thanks for your appreciation. The point-to-point responses are given as follows.

**Comment #2**

*There are many other reanalysis data available and why you choose the MERRA2*
*dataset? What is advantage of this dataset?*

**Authors' reply:**

Thanks for your comment. We agree with you that there are many other
reanalysis data, such as ERA-land (Muñoz-Sabater et al., 2021), GLDAS (Rodell et
al., 2004), JRA-55 (Kobayashi et al., 2015), and NCEP (Kalnay et al., 1996)
reanalysis datasets. We chose MERRA2 dataset because it can provide global hourly
air temperature. The MERRA2 air temperature can provide the annual air temperature
variation pattern to simulate LST fluctuations induced by synoptic conditions. This
information is used in the ATC model to reconstruct the under-cloud LSTs at four
overpassing times. Other reanalysis datasets can replace the MERRA2 dataset if they
could provide similar information.

**Comment #3**

*The key steps are suggested to be clarified in in figure 2. The pre-processing is not*
*included in this flowchart.*

**Authors' reply:**

Thanks for your comment. We have added the preprocessing steps which include
unifying the projection system and resampling the datasets to the same spatial
resolution in the flowchart. The revised flowchart is given as follows for your
convenience.

[Figure]

**Fig. 5. Flowchart of the IADTC framework. $DTR_{four}$ refers to diurnal temperature range (DTR) calculated as the maximum minus the minimum from the gap-free LSTs at the four overpassing times; $DTR_{DTC}$ refers to the DTR calculated from the hourly LSTs modelled with the DTC model. $\Delta DTR$ refers to the absolute difference between $DTR_{four}$ and $DTR_{DTC}$.**

**Comment #4**

*175: A basic equation of the single-type and multi-type model is better to be provided here.*

**Comment #5**

*Figure 3: multi-type ATC models are identical? Why there is no differences? It will be*

*a little confused on the naming of the ATC models for single or multi-type model and*
*single or double-sinusoidal ATC model?*

**Authors' reply:**

Thanks for your comment. Comments #4 and #5 are both related to descriptions
of ATC model, so we combine the response. We agree with you that some of the ATC
model descriptions are redundant and could be misleading.

We summarized the basic equation of ATC model as Eq. R1. For the single-type
ATC model, $M$ equals 1 for the global application, i.e., the single-sinusoidal version
was applied to the global scale. As for the multi-type ATC model, the value of $M$ is
different at different latitude zones. In low-latitude (23.5° N – 23.5° S) and high-
latitude regions (66.5° N/S – 90° N/S), $M$ equals 2, i.e., the double-sinusoidal version
was applied to these regions. In mid-latitude regions (23.5° N/S – 66.5° N/S), $M$
equals 1, i.e., single-sinusoidal version was used.

To address your question about the identical results between the single-type and
multi-type ATC models, the results of single-type and multi-type ATC models are
identical in mid-latitude region because they both use the single-sinusoidal version ($M$
= 1). Therefore, the results in Fig. 3b are identical. While the results of single-type
and multi-type ATC models are different in low-latitude and high-latitude regions
(Fig. 3a & Fig. 3c) because the single-type ATC model still uses the single-sinusoidal
version ($M$ =1) while the multi-type ATC model use the double-sinusoidal version ($M$
= 2).

$$\begin{cases} T_{\text{ATCM}}(d) = T_0 + \sum_{m=1}^{M} A_{\text{m}} \sin\left(\frac{2\pi m d}{N} + \theta_{\text{m}}\right) + k \cdot \Delta T_{\text{air}}(d) \\ \Delta T_{\text{air}}(d) = T_{\text{air}}(d) - T_{\text{ATCO}}(d) \\ T_{\text{ATCO}}(d) = T_0' + \sum_{m=1}^{n} A_{\text{m}}' \sin\left(\frac{2\pi m d}{N} + \theta_{\text{m}}'\right) \end{cases} \qquad \text{Eq. R1}$$

where $T_{\text{ATCM}}(d)$ denotes the daily LST variations simulated with the ATC model; $M$ is
the number of used harmonic components; $d$ and $N$ are the day of year (DOY) and
number of days in a year, respectively; $\Delta T_{\text{air}}(d)$ is the difference between the daily
SATs (i.e., $T_{\text{air}}(d)$, obtained from MERRA2 reanalysis data) and the modelled air
temperatures with the original ATC model ($T_{\text{ATCO}}(d)$); and $T_0$, $A_{\text{m}}$, $\theta_{\text{m}}$, and $k$ are the
parameters that need to be solved with the cloud-free daily LSTs and SATs, usually
through the least-square method.

To reduce the redundancy and clarify the description, we have revised Section
3.1.2. The revised version is given as follows for your convenience.

**3.1.2 Under-cloud LST reconstruction with multi-type ATC model**

The general formula of ATC model is displayed in Eq. 2. The single-type ATC model in the OADTC framework uses a single sinusoidal function ($M = 1$ in Eq. 2) to model the intra-annual LST variations driven by solar radiation change and incorporates surface air temperatures to help simulate the LST fluctuations induced by synoptic conditions (Zou et al., 2018; Liu et al., 2019b). The use of a single sinusoidal function is generally acceptable for mid-latitude regions. However, a single sinusoidal is no longer suitable for low-latitude because there are two solar radiation peaks within a yearly cycle over low-latitude regions (Xing et al., 2020; Bechtel, 2015; Cao and Sanchez-Azofeifa, 2017); it is also inadequate for high-latitude regions where polar days and nights occur (Østby et al., 2014; Liu et al., 2019; Westermann et al., 2012). Therefore, the use of the single-type ATC model in the OADTC framework is less suitable to generate $T_{dm}$ at the global scale (Fig. 6). To overcome this limitation, the IADTC framework uses different versions of ATC model (termed the multi-type ATC model) to reconstruct under-cloud LSTs over the low-, mid-, and high-latitude regions, respectively. The details are given as follows:

*(1) Low-latitude regions (23.5° N – 23.5° S)*

The solar radiation possesses two peaks within a yearly cycle over low-latitude regions (Fig. 6a). We therefore employed the ATC model with two sinusoidal functions *(M = 2 in Eq. 2)* to reconstruct the daily LST dynamics within an annual cycle (Liu et al., 2019b; Xing et al., 2020).

*(2) Mid-latitude regions (23.5° N/S – 66.5° N/S)*

The solar radiation peaks once in summer during an annual cycle. We therefore employed the ATC model with single-sinusoidal function *(M = 1 in Eq. 2)* to reconstruct the daily LST dynamics (Fig. 6b).

*(3) High-latitude regions (66.5° N/S – 90° N/S)*

The polar day/night phenomena occur over high-latitude regions and the duration increases with the latitude. Theoretically, over these regions, the ATC model with multiple sinusoidal functions should be the best choice. However, the number of cloud-free MODIS observations is limited, and additional model complexity can lead to over-fitting and weaken the generalization ability of the ATC model (Liu et al., 2019b). To balance model accuracy and generalization ability, the ATC model with two sinusoidal functions was selected for high-latitude regions (see Fig. 6c).

$$\begin{cases} T_{\text{ATCM}}(d) = T_0 + \sum_{m=1}^{M} A_{\text{m}} \sin\left(\frac{2\pi md}{N} + \theta_{\text{m}}\right) + k \cdot \Delta T_{\text{air}}(d) \\ \Delta T_{\text{air}}(d) = T_{\text{air}}(d) - T_{\text{ATCO}}(d) \\ T_{\text{ATCO}}(d) = T_0^{'} + \sum_{m=1}^{M} A_{\text{m}}^{'} \sin\left(\frac{2\pi md}{N} + \theta_{\text{m}}^{'}\right) \end{cases} \quad (2)$$

where $T_{\text{ATCM}}(d)$ denotes the daily LST variations simulated with the ATC model; $M$ is the number of used harmonic components; $d$ and $N$ are the day of year (DOY) and number of days in a year, respectively; $\Delta T_{\text{air}}(d)$ is the difference between the daily

SATs (i.e., $T_{\text{air}}(d)$, obtained from MERRA2 reanalysis data) and the modelled air temperatures with the original ATC model ($T_{\text{ATCO}}(d)$); and $T_0$, $A_{\text{m}}$, $\theta_{\text{m}}$, and $k$ are the parameters that need to be solved with the cloud-free daily LSTs and SATs, usually through the least-square method.

[Figure]

**Fig. 6. Comparison of reconstructing under-cloud LSTs with multi-type and single-type ATC models at different latitudes. (a), (b), and (c) show three examples of ATC modelling at low-, mid-, and high-latitudes for cloud-free Terra-day LST in 2019. The green circles, blue lines, and red lines denote the cloud-free observations and LSTs simulated by the single- and multi-type ATC models, respectively. Note that for (b) the results of the single- and multi-type ATC models are identical since they both use the ATC model with single-sinusoidal function.**

**Comment #6**

*Section 3.1.3: I think it should be the interpolation of the missing LSTs but not*

*overpassing times.*

**Authors' reply:**

Thanks for your comment. Section 3.1.2 is the under-cloud LST reconstruction and Section 3.1.3 is the interpolation of overpassing time. The interpolation of overpassing time is required because, in the original MODIS LST products (MOD11C1 and MYD11C1), not only the cloud contaminated LSTs are missing, but also the overpassing time of the cloud contaminated pixel. Because the overpassing time is synchronically masked with the cloud contaminated LST. The overpassing time is the required input variable in the DTC model, and the missing overpassing time cannot drive the DTC model. Therefore, we used linear interpolation to reconstruct the missing overpassing time, which is the content of Section 3.1.3.

**Comment #7**

*Actually, the DTC model should be not applied to get the DTCdm when there are*

*cloud-cover observations.*

**Authors' reply:**

Thanks for your comment. Although the current DTC model is designed for the clear-sky condition, it can be applied to estimate daily mean LST ($T_{dm}$) with acceptable accuracy. This has been validated by our previous study (Hong et al.,

2021). We acknowledge that under cloudy conditions, the DTC-modelled diurnal LST

dynamics (blue and red lines in Fig. R6) could have significant deviations compared with the actual diurnal LST dynamics (black line in Fig. R6). However, the aggregated $T_{dm}$ can still achieve satisfactory accuracy (Hong et al., 2021) because: (1)

the positive and negative biases of the modelled diurnal LST dynamic were partly offset when calculating the daily mean LST; (2) under cloudy condition, the diurnal

LST variation is relatively mild, which can also reduce the daily mean LST estimation error to some degree.

In this paper, we also validated the accuracy of $T_{dm}$ estimated with the DTC

model. For the SURFRAD datasets, the MAEs of estimated $T_{dm}$ at the daily and monthly scales are 1.4 K and 0.6 K, respectively (Fig. 6). For the FLUXNET datasets, the MAEs of $T_{dm}$ are 1.1 K and 0.5 K at the daily and monthly scales, respectively (Fig. 7). The validation results show that the DTC model can be applied to estimate daily mean LST under cloudy conditions.

[Figure]

Fig. 12. Typical examples of DTC modelling results obtained for six SURFRAD sites in 2017. The blue (red) numbers in the upper right corners provide the MAEs of $T_{in\_ATC\_DTC}$ and $T_{dm\_ATC\_DTC}$ ($T_{in\_obs\_DTC}$ and $T_{dm\_obs\_DTC}$). In (a), the conditions are completely cloud-free: therefore, the results for $T_{in\_ATC\_DTC}$ and $T_{in\_obs\_DTC}$ are identical (i.e., ATC modelling is not needed). (b)-(f) represent the cases with increasing cloud contamination. (For interpretation of the references to colour in this figure legend, the reader is referred to the web version of this article.)

Fig. R6. Screenshot of Fig. 12 in Hong et al. (2021).

**Comment #8**

*Besides the direct validation of the estimated mean values at different temporal scales,*

*there is a lack of the evaluation of the reliability of the trend detection based on the*

*generated dataset. How about the performance of the dataset on identifying the area*

*with significant trends.*

**Authors' reply:**

**34 / 42**

Thanks for your comment. To evaluate the reliability of the LST trend based on the generated daily mean LST, ground truth is required. The LST trend calculated based on the *in situ* measurement is sensitive to the local climate variation, and there is a scale mismatch between the site-level LST trend and pixel-level LST trend. Therefore, the LST trend based on the *in situ* measurement might not be representative to evaluate the LST trend based on the generated daily mean LST dataset.

Acquiring the ground truth to validate the generated daily mean LST product could be costly and complicated. Consequently, to evaluate the reliability of the LST trend detection based on the generated GADTC dataset, we compare the LST trend based on generated GADTC products with other studies. We found that the LST trend detected based on the generated GADTC products (Fig. 10) is similar to the previous studies conducted by Sobrino et al. (2020) (Fig. R7) and Mao et al. (2017) (Fig. R8). Additionally, we provided the LST anomalies from 2003 to 2019 of each continent and global scale (Fig. R9). Fig. 10 and Fig. R9 both confirm the significant trends in certain areas, such as the warming and Europe and Arctic.

[Figure]

[Figure]

**Figure 4.** Global linear trend map for the period 2003–2016 estimated by the linear (**left**) and Sen's slope (**right**) methods, with results of 0.018 °C/yr and 0.017 °C/yr, respectively. The Mann-Kendall test significance map is also provided.

Fig. R7. Screenshot of Figure 4 in Sobrino et al. (2020) describing the global LST

trend.

[Figure]

Fig. 5. Global surface temperature change from 2001 to 2012: (a) rate (slope) of linear regression and (b) correlation coefficient.

Fig. R8. Screenshot of Figure 5 in Mao et al. (2017) describing the global LST trend.

[Figure]

Fig. R9 LST anomalies as well as the associated linear regressions for $T_{dm\_cloud\_free}$ and $T_{dm\_IADTC}$ from 2003 to 2019. (**a**) displays the global LST anomalies; and (**b**) to (**h**) display the LST anomalies for each continent.

**Comment #9**

*The threshold determination for the two criteria in Fig. 2 is a little objective. I think the determination can be automatically determined according to the differences between the average value from four observations and the fitted values.*

**Authors' reply:**

Thanks for your comment. Actually, we tried automatically determining the threshold according to the average value from four observations (i.e., $T_{\text{dm\_ATC\_four}}$) and the DTC-fitted values (i.e., $T_{\text{dm\_ATC\_DTC}}$) when constructing the IADTC framework. We found it hard to design a concise rule to automatically differentiate different scenarios based on the difference between $T_{\text{dm\_ATC\_four}}$ and $T_{\text{dm\_ATC\_DTC}}$. Therefore, we remain choosing to use the fixed threshold.

We agree with you that there are other strategies to determine the thresholds. Those strategies might achieve better accuracies. However, our current validation results show that simply using the fixed threshold can already achieve satisfactory accuracy.

**Comment #10**

*The LSTs of cloud cover pixels are generated with the reanalysis data at coarse-resolution. Currently, there are some other reconstruction methods without the use of the reanalysis data. How about the applicability of these methods in this study.*

**Authors' reply:**

Thanks for your comment. The role of ATC model is to reconstruct the under-cloud LST with the assistance of reanalysis data. There are some other reconstruction methods without using the reanalysis data, such as statistical interpolation, spatiotemporal fusion, and passive microwave-based method (Wu et al., 2021; Hong et al., 2021). Additionally, previous studies have produced seamless LST datasets (Zhang et al., 2022; Zhao et al., 2020). These methods or products can replace the ATC model in our $T_{\text{dm}}$ generation framework. We have clarified this point in Line 547-551, which was given as follows for your convenience.

Line 547-551:

{Third, other high-efficient under-cloud LST reconstruction methods, such as statistical interpolation, spatiotemporal fusion, and passive microwave-based method (Wu et al., 2021; Hong et al., 2021), or the generated under-cloud LST products (Zhang et al., 2022; Zhao et al., 2020), can replace the ATC model in the $T_{\text{dm}}$ generation framework. Similarly, more efficient diurnal LST dynamics modelling methods can also replace the DTC model (Jia et al., 2022).}

**Comment #11**

*The dataset produced in this study has the resolution of 0.5 degree. However, to some extent, the LST product at 1-km and higher resolution will be useful. What is the key issue should be addressed at this high-resolution level.*

**Authors' reply:**

Thanks for your comment. We agree with you that 1-km or higher resolution LST products are useful and valuable. Our IADTC framework can be directly applied to the 1-km MODIS LST to generate $T_{dm}$ in a small region. Our previous study provides the example of generating 1-km $T_{dm}$ in Shanghai using the OADTC framework. It can also be generated using the IADTC framework. You can refer to Fig. S1 in (Hong et al., 2021) for more details.

While for generating long-term and large-scale 1-km resolution LST product, calculation efficiency and computation complexity is the key issue. The tons of DTC model fitting using the least-square fitting cover the majority of running time. In the future perspective section, we mentioned three possible ways to reduce the computation complexity and improve the calculation efficiency. The first is to use the similarity of the ATC and DTC model parameters among neighboring pixels to reduce the computation complexity. The second is to combine statistical or empirical estimation strategies to reduce the times of least-square fitting and improve computational efficiency. The third is to use other high-efficient methods to replace the ATC or DTC model in the $T_{dm}$ generation framework. We have provided elaborated descriptions about this point in Line 530-551, which were given as follows for your convenience.

[revised manuscript text omitted]

Østby, T. I., Schuler, T. V., and Westermann, S.: Severe cloud contamination of MODIS Land Surface Temperatures over an Arctic ice cap, Svalbard, Remote Sens. Environ., 142, 95-102, doi:10.1016/j.rse.2013.11.005, 2014.

Rodell, M., Houser, P. R., Jambor, U., Gottschalck, J., Mitchell, K., Meng, C.-J., Arsenault, K., Cosgrove, B., Radakovich, J., Bosilovich, M., Entin, J. K., Walker, J. P., Lohmann, D., and Toll, D.: The Global Land Data Assimilation System, Bull. Am. Meteorol. Soc., 85, 381-394, doi:10.1175/BAMS-85-3-381, 2004.

Sobrino, J. A., Julien, Y., and García-Monteiro, S.: Surface temperature of the planet Earth from satellite data, Remote Sens., 12, 218, doi:10.3390/rs12020218, 2020.

Wan, Z. and Li, Z. L.: A physics-based algorithm for retrieving land-surface emissivity and temperature from EOS/MODIS data, IEEE Trans. Geosci. Remote Sens., 35, 980-996, doi:10.1109/36.602541, 1997.

Westermann, S., Langer, M., and Boike, J.: Systematic bias of average winter-time land surface temperatures inferred from MODIS at a site on Svalbard, Norway, Remote Sens. Environ., 118, 162-167, doi:10.1016/j.rse.2011.10.025, 2012.

Wu, P., Yin, Z., Zeng, C., Duan, S.-B., Gottsche, F.-M., Ma, X., Li, X., Yang, H., and Shen, H.: Spatially continuous and high-resolution land surface temperature product generation: A review of reconstruction and spatiotemporal fusion techniques, IEEE Geoscience and Remote Sensing Magazine, 9, 112-137, doi:10.1109/mgrs.2021.3050782, 2021.

Zhang, T., Zhou, Y., Zhu, Z., Li, X., and Asrar, G. R.: A global seamless 1 km resolution daily land surface temperature dataset (2003–2020), Earth System Science Data, 14, 651-664, doi:10.5194/essd-14-651-2022, 2022.

Zhang, X., Zhou, J., Liang, S., Chai, L., Wang, D., and Liu, J.: Estimation of 1-km all-weather remotely sensed land surface temperature based on reconstructed spatial-seamless satellite passive microwave brightness temperature and thermal infrared data, ISPRS J. Photogramm. Remote Sens., 167, 321-344, doi:10.1016/j.isprsjprs.2020.07.014, 2020.

Zhao, B., Mao, K., Cai, Y., Shi, J., Li, Z., Qin, Z., Meng, X., Shen, X., and Guo, Z.: A combined Terra and Aqua MODIS land surface temperature and meteorological station data product for China from 2003 to 2017, Earth System Science Data, 12, 2555-2577, doi:10.5194/essd-12-2555-2020, 2020.